# “Omic” Approaches to Bacteria and Antibiotic Resistance Identification

**DOI:** 10.3390/ijms23179601

**Published:** 2022-08-24

**Authors:** Daria Janiszewska, Małgorzata Szultka-Młyńska, Paweł Pomastowski, Bogusław Buszewski

**Affiliations:** 1Department of Environmental Chemistry and Bioanalytics, Faculty of Chemistry, Nicolaus Copernicus University, Gagarina 7, 87-100 Torun, Poland; 2Centre for Modern Interdisciplinary Technologies, Nicolaus Copernicus University, Wilenska 4, 87-100 Torun, Poland

**Keywords:** antibiotic resistance, bacteria identification, MALDI-TOF MS, identification methods, “omic” technique

## Abstract

The quick and accurate identification of microorganisms and the study of resistance to antibiotics is crucial in the economic and industrial fields along with medicine. One of the fastest-growing identification methods is the spectrometric approach consisting in the matrix-assisted laser ionization/desorption using a time-of-flight analyzer (MALDI-TOF MS), which has many advantages over conventional methods for the determination of microorganisms presented. Thanks to the use of a multiomic approach in the MALDI-TOF MS analysis, it is possible to obtain a broad spectrum of data allowing the identification of microorganisms, understanding their interactions and the analysis of antibiotic resistance mechanisms. In addition, the literature data indicate the possibility of a significant reduction in the time of the sample preparation and analysis time, which will enable a faster initiation of the treatment of patients. However, it is still necessary to improve the process of identifying and supplementing the existing databases along with creating new ones. This review summarizes the use of “-omics” approaches in the MALDI TOF MS analysis, including in bacterial identification and antibiotic resistance mechanisms analysis.

## 1. Introduction

Multi-drug resistant bacterial infections disease cause 700,000 deaths globally every year. It is estimated that this number could increase to 10 million by 2050 [1,2]. According to the statistical data, multi-drug resistant strains have quadrupled worldwide. The recent work by Cassini et al. emphasizes that antibiotic resistance is currently one of the most significant public health challenges. It indicates a strong influence of resistant pathogens on the incidence of clinical infections [3]. The development of fast, accurate, and sensitive methods of identifying pathogenic clinical bacteria is essential for the correct microbiological diagnostics and the implementation of the individual antibiotic therapy. In addition, it reduces the chances of the development of multi-drug resistance [4]. Monitoring biological hazards in the environment and detecting pathogens in foodstuffs, such as *Salmonella* [5], *Listeria monocytogenes* [6], *Bacillus cereus* [7], *Campylobacter jejuni* and *Clostridium perfringens* [8] are also crucial to protect human health. New antibiotics and therapies are urgently needed to control these infections, while new rapid and reliable diagnostic techniques are necessary to characterize strains.

Bacterial infections may lead to the development of diseases that are dangerous to life and health. They include diabetic feet [9], sepsis [10] or gonorrhea [11]. Various techniques are routinely used in clinical microbiology, including biochemical, serological, and chemotaxonomic molecular biology techniques. However, these methods are labor-intensive, time-consuming (lasting up to 3 days), and often inadequate to differentiate phenotypically similar species, particularly when using spectroscopic and spectrometric tools.

Mass spectrometry (MS) was developed in the late 19th century to measure the masses of atoms. The MS technique is an analytical approach used to measure the mass to charge (*m*/*z*) ratio of chemicals and to calculate their exact mass. Currently, the most common ionization techniques used to analyze chemical structures in biological systems are the laser desorption/ionization (LDI), matrix-assisted laser desorption ionization (MALDI), or surface-enhanced laser desorption/ionization (SELDI) and electrospray ionization (ESI). It is becoming increasingly common to combine the MALDI-TOF MS with “-omics” approaches that take a holistic view of the molecules making up an organism. They are primarily aimed at the global detection of genes (genomics), mRNAs (transcriptomics), proteins (proteomics), lipids (lipidomics) and metabolites (metabolomics) in a biological sample.

## 2. MALDI-TOF MS Technique

The Matrix-Assisted Laser Desorption/Ionization-Time of Flight (MALDI-TOF) mass spectrometry (MS) method was developed in the 1980s. The name “MALDI” was first used in 1985 by Hillenkam et al. [12]. At the same time, the work on the MALDI technique and the possibility of its application to the protein analysis was conducted by Tanaka, for which he was awarded the Nobel Prize in 2002 [13]. In the late 1990s, the pioneering use of MS in microbiology showed that intact bacterial cells could be distinguished using the MALDI coupled to a time-of-flight (TOF) analyzer [14].

MALDI is a soft ionization technique in which samples are ionized to charge molecules, while their *m*/*z* can be measured. In this method, the analyte is mixed with a small molecule compound known as the matrix before measurement, which mediates the energy transfer to the test substance, facilitates the sample ionization and allows conducting the study of non-volatile, high molecular weight and polar substances. The matrices in the MALDI technique are compounds that absorb UV radiation well, quickly sublimate and, after the desorption process, provide large amounts of ions (protons in a positive mode or anions in a negative mode) needed for the ionization of the test substance [5]. The principle of the technique is to deposit the analyte in a huge excess of the matrix compound deposited on a solid surface called a target, usually made of conductive metal and having spots for several different samples to be applied. After a very short laser pulse, the irradiated spot heats up quickly and becomes vibration-excited. The matrix particles energetically removed from the sample surface absorb the laser energy and transfer the analyte particles to the gas phase. During the ablation process, analyte molecules are usually ionized by protonation (positive ion mode [M + H]^+^) or deprotonation (negative ion mode [M−H]^−^) with nearby matrix molecules. The most common format for MALDI ionization is that analyte molecules carry a single positive charge [15]. Molecular weight is measured by mass spectrometry, and the detector calculates the time of flight of the ions (TOF). The basic principle of TOF is that ions of different *m*/*z* are time-scattered as they fly along a field-less drift path of known length. Assuming all the ions begin their transfer simultaneously, or at least in a short enough time interval, the lighter ions will reach the detector sooner than the heavier ones [16]. The final result of the analysis is the spectrometric spectrum, which shows the masses of the formed ions and non-ionized molecules. The signals are arranged according to the increase in mass.

The selection of an appropriate matrix is one of the key steps in the sample preparation protocol for the analysis. Examples of the matrices used in the MALDI-TOF MS technique are shown in Figure 1.

The most commonly used matrices in the MALDI-TOF MS analysis are 2,5-dihydroxybenzoic acid (gentisic acid, DHB), 4-hydroxycinnamic acid (sinapinic acid, SA), and α-cyano-4-hydroxycinnamic acid (HCCA). The DHB matrix enables the analysis of oligosaccharides, glycopeptides and glycoproteins. DHB is more efficient for low-molecular-weight molecules, while SA and HCCA are especially used for protein studies [17]. The application of ferulic acid (FA) allows for proteins to be tested with a molecular weight of up to 70 kDa. For the same species, the mass spectrum fingerprints differ depending on the matrices used, which underlines the need for the careful selection of the appropriate matrix [18]. Despite many advantages of the HCCA matrix, such as better sensitivity and a greater number of signals at the lower mass limits, compared to FA and SA, HCCA shows a lower signal resolution and an increased degree of peak broadening. Moreover, the spectra generated with HCCA also lack mass signals as compared to e.g., ferulic acid. Fagerquis et al. reported that the use of sinapinic acid (SA) revealed more signals on the mass profile of *E. coli* bacterial cell lysates than in the case of HCCA [19]. Additional peaks appear at *m*/*z* ~208 greater than the *m*/*z* of the more abundant protein ion peak. Importantly, only those proteins containing cysteine showed reactivity with SA. Šedo et al. used a new protocol in which they used FA instead of HCCA as a template. They proved that the new protocol allowed them to extend the range of detected compounds towards a higher molecular weight and to generate signals with better mass resolution. As a result, the differentiation of *Acinetobacter nosocomialis* and *Acinetobacter baumannii* strains was improved, while *A. nosocomialis* strains, incorrectly or ambiguously assigned using the standard protocol, were correctly identified [20]. The main disadvantage of using FA as a matrix, however, is the lack of the automatic acquisition of the mass spectra because it gives a heterogeneous layer of matrix crystals.

The molecules 2-mercaptobenzothiazole (MBT) or 5-Chloro-2-mercaptobenzothiazoles (CMBT) are used as a matrix for the ionization of lipids or phospholipids that make up cell walls and membranes (such as from intact Gram-positive bacterial cells or *Bacillus* spores). Scientists use several matrices to analyze lipid A, including DHB, 2,4,6-trihydroxyacetophenone (THAP) and 6-aza-2-thiotimine (ATT) [21]. Although DHB is widely used for peptide analyses, it produces uneven crystals and leads to intra-sample variability (most commonly referred to as point-to-point reproducibility). In addition, low solubility in the lipid A compatible solvent and heterogeneity in the matrix layer (crystals) may lead to changes in the ionization efficiency throughout the sample [22]. Shu et al. investigated the lipid profiles of *Bacillus* spp. spores using the MALDI-TOF MS in silico. They analyzed several matrices: 2,5-DHB, 2,4-DHB, SA, HCCA, THAP and 2-(4-Hydroxyphenylazo)benzoicacid(HABA). They showed that the matrix had no significant effect on the lipid analysis apart from the signal intensity. The mass spectra obtained from 2,5-DHB and HCCA had a lower signal-to-noise ratio and lower signal intensity. This may be due to the lower absorption coefficient of the matrices at the laser wavelength of 266 nm. HABA produced more matrix cluster ions with mass peaks above 900 *m*/*z*. These cluster ions can overlap with the mass peaks of the lipids. Therefore, HABA is not suitable for lipid ionization. A similar matrix effect phenomenon was observed in other bacterial mass spectra besides those obtained for 2,4-DHB and THAP, which are better matrices for lipid analyses at a laser wavelength of 266 nm and in a positive mode [23].

Xu et al. proved that CMBT provides excellent point-to-point repeatability due to the homogeneous crystallization of the analyte/matrix mixture over the sample point [24]. Moreover, CMBT is a soluble solvent compatible with the lipid A molecules and, therefore, is widely used for lipid A analysis [25,26]. Liu et al. on the other hand, studied the effect of the HCCA and CMBT matrix on the identification level and quality of the mass spectra of *Yersinia pestis*, *Escherichia. coli*, *Burkholderia cepacia*, *Bacillus anthracis*, and *Staphylococcus aureus* bacteria. They showed that the best signals were obtained using CMBT. The HCCA matrix has a higher chemical activity than CMBT and is therefore susceptible to reactions with other reagents, which can result in a greater ionization effect and susceptibility to contaminants present in the sample [27]. Elhanany et al. compared HCCA and SA in the analysis of intact *B. cereus* group bacterial spores and observed that HCCA tends to give mass spectra in which characteristic molecules of lower masses are more pronounced, whereas SA seems to be more suitable for those of high masses. This finding was also supported by Horneffer et al. [28,29]. For the fungal spore analysis, inconsistent results were described regarding the effect of matrices on the mass spectrum. Armiri et al. compared SA, HCCA, FA, HPA and DHB, concluding that the use of SA allowed for obtaining the best quality spectra, and DHB the worst [30], while Valentine et al. found that ferulic acid enabled better results than SA [31]. Li et al., who investigated the possibility of using SA and HCCA, proved that both matrices were equally suitable for the analysis of fungal spores [32]. Similar inconsistent observations are described for the analysis of whole bacterial cells [33,34,35]. It was also observed that different batches, i.e., different suppliers and lots, of the same matrix substance lead to different peak patterns [28]. Nevertheless, these results demonstrate that the matrix selection for the MALDI-MS analysis of complex characteristic molecules is challenging.

Moreover, MALDI-TOF MS works in the positive and negative ion mode, along with the linear and reflectron mode, thus achieving four combinations: LP (linear positive), LN (linear negative), RP (reflectron positive), RN (reflectron negative). The key difference between positive and negative ionization in mass spectrometry is that the positive ionization is the process which leads to the production of positively charged ions, while the negative ionization is the process through which negatively charged ions are generated [36]. The workflow in the reflectron mode is almost the same as in the linear mode. The only difference is that when an ion hits the reflector, it will bounce and fly towards the detector. The reflectron focuses ions with the same *m*/*z* values and makes them reach the detector at the same time, resulting in a more accurate detection. By using different MALDI-TOF MS modes, a greater variety of results can be obtained from a single sample analysis [37,38,39].

The main advantage of the MALDI-TOF MS technique is that the identification time of microorganisms is significantly reduced by 24 to 36 h when compared to conventional techniques, and the analysis time is 5.1 min of hands-on time/identification. Another advantage of MALDI-TOF in microbiological diagnostics is the low unit cost of analysis ($0.50/sample) [40]. In addition, the use of “soft” ionization in MALDI-TOF allows for the observation of ionized molecules with little or no fragmentation because the resulting ions have a low internal energy. The most significant limitation of MALDI-TOF is its low analytical sensitivity without prior cultivation and the discrimination of phyletically related microorganisms such as *Shigella* and *Escherichia coli* [41] Consequently, MALDI-TOF is unsuitable for detecting the small number of bacteria in sterile samples. Also, the initial equipment basket, around $270,000, is a non-tape disadvantage [42]. Currently, it is also impossible to sequence individual bacterial proteins directly during the analysis in a linear mode.

Moreover, there are modifications to the MALDI technique, such as matrix-assisted laser desorption/ionization Fourier transform mass spectrometry (MALDI-FTMS), matrix-assisted laser desorption/ionization Fourier transform ion cyclotron resonance mass spectrometry (MALDI-FT-ICR) or MALDI mass spectrometry imaging (MALDI-MSI) allowing for the analysis of various types of molecules with often more excellent resolution and accuracy [43,44,45]. The techniques listed are described in more detail later in the manuscript.

## 3. Bacteria Identification

### 3.1. Proteomic

The proteome is the entire set of proteins present in a cell at any given time. Proteomics refers to the experimental analysis of proteins and proteomes, which often entails the protein purification and the mass spectrometry analysis. The amino acid sequences of many proteins differ between microorganisms, and many different analytical techniques are used to characterize and differentiate microorganisms based on the proteome analysis.

The gold standard in the identification and classification of bacteria is the sequencing of the gene encoding 16S rRNA (16S rDNA) due to its high conservation within species, inter-species variability and the stable rate of evolution. The 16S rRNA gene is universal in bacteria so that relationships between all bacteria can be measured, and the comparison of gene sequences enables the differentiation between microorganisms on the genus level and the classification of whispers on the species and subspecies levels.

The MALDI-TOF MS method detects proteins ranging in mass from 2 to 20 kDa, which mainly represent ribosomal proteins and essential metabolism proteins. Ribosomal proteins are among the most conservative proteins in all life forms [46]. The conservativeness of the ribosomal proteins and fragments of the 16S rRNA gene enable the use of the proteomic MALDI-TOF MS and sequencing to obtain comparable identification results [47]. The analysis of ribosomal proteins by means of MALDI-TOF MS allows profiling “fingerprints” that are characteristic of a microorganism and comparing the protein profile with the library of reference spectra. This, in turn, allows for the taxonomic position of the microorganism to be determined according to the level of the genus and, in many cases, also to the species or even strain [48]. The advantage of the MALDI-TOF MS technique over the 16S rDNA sequencing consists of the possibility of not only identification, but also the analysis of the response to environmental conditions, including the antibiotics used, and thus the detection of antibiotic resistance [41].

During the MALDI-TOF analysis, two parameters are assessed for each ion: the mass-to-charge ratio (*m*/*z*) and the relative intensity of the ion. The identification of microorganisms by means of MALDI-TOF MS is carried out by comparing the protein profile of an unknown organism with the reference profiles contained in the library. Depending on the degree of the similarity of the obtained spectrum and the reference spectrum, the microorganism is identified to the level of genus, species, subspecies or strain [49]. The procedure for analyzing bacterial proteins by MALDI-TOF MS assumes that using one colony (10^4^–10^5^ CFU/spot) of the tested microorganism is sufficient to obtain mass spectra. In the case of tiny colonies, this is not possible. The study must be retrieved by several colonies [50].

The MALDI-TOF MS analysis of bacterial proteins is possible using one of two strategies: the “bottom-up” methods for peptide mixtures derived from protein digestion (i.e., peptide sequencing) and the “top-down” method for the direct analysis of intact proteins, proteoforms and post-translational protein modifications [51]. The bottom up approach allows for the identification of proteins along with the characterization of amino acid sequences and post-translational modifications. This approach, however, requires the proteolytic digestion of the peptides and often a pre-fractionation step. The digested samples are then analyzed by spectrometry. Differential expression using the bottom up approach often involves labeling the sample with isobaric tags prior to digestion. All bottom-up methods require high-resolution and high-performance instrumentation [52,53]. Dickinson et al. attempted to identify characteristic proteins in the MALDI-TOF MS profiles of *B. subtilis* and noted that assigning the appropriate signals to specific proteins was difficult. They also found that the separation of proteins, prior to using the bottom-up approach, was necessary to increase confidence in linking the identified proteins to signals observed in the MALDI-TOF MS profiles generated from intact bacterial cells [54]. Faqerusta et al. used HPLC and one-dimensional sodium dodecyl sulfate polyacrylamide gel electrophoresis (1D SDS-PAGE) [55] to separate *Compylobacter* sp. protein extracts before identification. Schaller et al. also pre-fractionated proteins prior to identification, but they used two-dimensional gel electrophoresis (2D GE) instead of the simpler 1D approach [56]. A similar labor-intensive approach to profiling *Lactobacillus plantarum* cultures at the strain level with 2D GE and the Peptide Mass Fingerprinting technique (PMF) was described by Sun et al. [57]. Additionally, Schmidt et al. differentiated *Lactobacillus* strains, where they used the trypsin digestion of cells from *Lactobacillus* reference strains and strains from dental patients’ teeth. The resulting digestion products were profiled using MALDI-TOF MS and a mass spectral library was created to categorize the unknown strains into their respective subspecies [58]. In contrast, Camara and Hays initially fractionated proteins using 1D SDS-PAGE and identified a PMF protein to confirm the ampicillin resistance (β-lactamase) in an ampicillin-resistant *E. coli* strain [59]. Unlike the bottom-up methods, in which the discovery of specific proteins is based on more specific and limited sample sets, the starting point for top-down proteomics can be hundreds of different complex biological samples. Researchers using top-down approaches are generally interested in solving clinical problems that require a larger number of samples, for example, biomarker discovery using body fluids such as blood, urine, plasma or saliva [60].

#### 3.1.1. Database

In microbiological laboratories, there are mainly two MALDI-TOF systems that analyze intact proteins: microflex^®^ LT/SH MS or Biotyper (Bruker Daltonics GmbH, Bremen, Germany) and VITEK^®^ MS (bioMérieux, Marcy l’Etoile, France) [61]. Both systems are available in the Research Use Only (RUO) and in vitro Diagnostic (IVD) versions. In each case, the detection range of the TOF analyzer is quite similar, but each is based on its own sets and databases [62,63]. Species identification with the commonly used MALDI-TOF MS systems is based on comparing unknown spectra with spectral reference databases through pattern matching. The MALDI-TOF mass spectra consist of peaks from many intracellular proteins, including ribosomal subunit proteins present in high copy numbers in replicating bacterial cells.

The MALDI Biotyper uses a pattern matching approach [3] with a database containing references referred to as the Main Spectrum Profile (MSP). The similarity of the obtained and reference spectra is expressed as “log (scoring)” where the value ≥2.3 means “high confidence identification”, between 2.0 to 2.3 means “secure genus identification”, ≥1.7 and <2 means “low confidence identification”, and a score < 1.7 is interpreted as “no reliable identification”. Besides this, the consistency of the 10 best results is another parameter to evaluate the identification. In the case of *Mycobacteria*, the following thresholds are adopted: ≥1.8—high confidence level and ≥1.6—low confidence level [64]. The described approach is the same for both RUO and IVD versions. The VITEK^®^ MS system with the IVD version uses an algorithm based on machine learning, “Advanced Spectra Classifier”. Spectra between 3000 and 17,000 Da are divided into 13,000 segments and then weighted according to their importance for identifying a given bacterial species. Unknown spectra undergo the same process to be compared successively with the Vitek MS database. The obtained results are given in percentages: 99.9%—perfect match, from 60% to 99.8%—good match, while values <60% are considered as no identification. In the SARAMIS system (RUO by Vitek MS), matching is calculated on the basis of typical strains that include intraspecific species diversity. The identification of unknown strains is made by a comparison with the spectra in the “SuperSpectra” database, and the confidence levels are given from high (>98%) to medium (85% to 98%) to low (75% to 85%) [65]. The research shows that both systems reveal similar identification rates [66,67].

MALDI-TOF MS analysis based on the database matching algorithm relies on fewer spectral attributes, such as the area under the peak and peak height, which are related to microbial species [68]. Consequently, there is much information contained in MALDI-TOF MS that remains untapped. Machine learning (ML) is a group of methods for finding patterns from specific datasets. Various ML algorithms, including k-nearest neighbors (KNN), naive Bayes (NB), random forest (RF) and support vector machine (SVM), are stable and reliable [69]. The ML model can use a large number of computations to discover non-intuitive or even counterintuitive statistical information from the learning set and use the learned pattern to classify the unknown test set [70]. In a recent paper, Weis et al. analyzed 36 studies implementing machine learning algorithms. These studies investigated bacterial species identification and antimicrobial susceptibility testing using MALDI-TOF MS. It is interesting to note that the vast majority of these studies used off-the-shelf classification methods in combination with relatively small datasets, usually containing less than a thousand samples and a minimal number of species, which are often restricted to a single family or even a single genus [68]. Mortier et al. conducted a large-scale comparative study of bacterial identification using MALDI-TOF mass spectrometry and machine learning methods. They implemented several traditional machine learning methods and several novel methods such as univariate conventional neural networks, hierarchical classifiers, and an out-of-decomposition detection method to identify *Leuconostoc* and *Fructobacillus* species. The results show that acceptable identification rates were obtained, but these numbers are typically lower than reported in studies with more limited analyses. Using hierarchical classification methods, researchers also showed that taxonomic information is generally not well preserved in MALDI-TOF mass spectrometry data [71].

Hyeon Park et al. compared the identification performance of the recently developed Autof ms1000 (Autobio Diagnostics Co., Ltd., Zhengzhou, China) with that of the Bruker Biotyper (Bruker Daltonics GmbH, Bremen, Germany). Studies reveal that both instruments showed comparable performance in the routine identification of clinical microorganisms [72]. Buchan et al., used the Mycobacterium Library v1.0 as an addition to the Mycobacterium specific spectral library used with the standard MALDI Biotyper software to identify the mycobacterium bacteria. The percentage of isolates generating an acceptable confidence result (≥1.7) increased from 50.6% (Biotyper standard library) to 89.8% (79/88) using the Mycobacterium Library v1.0 [73]. Farfour et al. analyzed a large pool of Gram-positive *Mycobacteria* using the Andromas system and reported the accurate identification of GPR species using the direct transfer with the additional ethanol treatment to fix and inactivate microorganisms [74]. The Andromas identification strategy is based on a limited number of species-specific profiles for each entry [75,76]. The unique feature of the Andromas database is that it was built without any extraction step. Regoui et al., on the other hand, developed a database to identify *Francisella*
*tularensis* and distinguished it from the closely related species *F. tularensis* subsp. *novicida* and *Francisella philomiragia* [77]. They also found that incubation on chocolate agar plates supplemented with PolyViteX^®^ at 30 or 37 °C in an aerobic or 5% CO_2_-enriched atmosphere for less than 72 h allows for the accurate identification of *F. tularensis* subsp. *holarctic*. Korean scientists developed a new MALDI-TOF MS ASTA MicroIDSys system (ASTA, Suwon, Korea). Compared to 16S rRNA sequencing and the Bruker Biotyper system, the ASTA MicroIDSys showed excellent results in identifying clinically significant anaerobic bacteria such as *Peptostreptococcus*
*anaerobius* [78], *Clostridium difficile*, *Clostridium perfringens*, *Finegoldia magna*, *Parvimonas micra* [79] and aerobic *Mycobacterium* [80]. The Autof MS 1000ASTA and MicroIDSys systems are analogous to the MALDI BioTyper, the database is based on an isolate-specific references approach, while forbioMérieux principles (e.g., Vitek MS) are based on taxonomical group-specific principles [81,82].

Another technique used to identify microorganisms is the approach that uses signals conserved from specific proteins found in bacterial cells. Ribosomal proteins proved to be one of the best biomarkers, because they are numerous, highly conserved and encoded by chromosomal genes. Their molecular weights range from 4 to 30 kDa, observed through MALDI-TOF MS [83]. Despite being highly conserved, interspecies and interstrain differences can be used in the typing and subtyping of microorganisms. Reference databases containing predicted weights of bacterial ribosomal subunits calculated directly from genomic sequences became an alternative to the pattern-based identification of bacteria in MALDI-TOF MS. A database PAPMID™ (Mabritec AG, Riehen, Switzerland) of putative protein masses for the identification was established, which was shown to complement reference databases such as SARAMIS™ (Mabritec, Riehen, Switzerland) [83]. Suarez et al., on the basis of ribosomal signals, grouped the different strains of *Neisseria meningitidis* into six subgroups corresponding to sequence types [84]. This approach in the MALDI-TOF MS analysis was successfully used to distinguish subspecies and clone complexes of bacteria such as *Streptococcus agalactiae* [85] and *E. coli* [86]. This method was also used successfully by Toh et al. for the differentiation of *Acinetobacter haemolyticus* and *Acinetobacter* genomic species, including 13BJ/14T strains [87].

#### 3.1.2. Sample Preparation

The key element in each “omic” approaches is the sample preparation stage. The methods of sample preparation to identify microorganisms using MALDI-TOF MS on the basis of ribosomal proteins include the preparation of ethanol–formic acid protein extracts, direct transfer and direct transfer with formic acid (Figure 2). Ethanol–formic acid extraction is the gold standard used to generate a reference database [88]. Schulthess et al. compared the three methods mentioned above to identify Gram-positive rods. The mass spectra analysis was performed by scientists using a Microflex LT mass spectrometer (Bruker Daltonics GmbH, Bremen, Germany). Their research showed that the identification rates for the direct formic acid transfer method were comparable to those of the ethanol–formic acid extraction procedure [89]. However, the protein extraction method is time-consuming [90]. Therefore, the direct transfer method is more effective in the routine clinical analysis. For many environmental strains such as *Legionella* spp., the direct sample transfer compared to the extraction procedure has no significant differences in the identification levels [91]. Direct sample transfer gives the best identification results for rod-shaped Gram-negative bacteria [92]. Worse outcomes were obtained for anaerobic bacteria, Gram-positive bacteria and some *Mycobacteria*. Studies show that *Bacillus subtilis* was misidentified as *Bacillus mojavensis* and vice versa, which may be due to the high similarity of the mass spectra of the two bacteria, which leads to eventual misidentification [93,94]. Gram-positive bacteria with a thick cell wall have a greater range of identification results, but are not always identified to the species level. In the case of these strains, it is difficult to obtain a smear that contains an adequate number of bacterial cells and is homogeneous [93,95]. To conclude, the extraction methods, due to higher protein recovery, are more preferable for MALDI detection of Gram-positive bacteria (in particular, sporulating bacteria).

Rotcheewaphan et al. developed a one-step method of extracting proteins from *Mycobacteria* using only a 1 µL loop of bacteria. Thus, they shortened the sample preparation time from 60 min to less than 10 min, ensuring clinically acceptable identification results (score > 1.8) [96]. The high level of identification caused that the application of bacterial colonies directly to the MALDI-TOF MS target plate became a standard protocol for the sample preparation in routine diagnostics [97].

In the case of the direct analysis of liquid clinical specimens, it is necessary to pre-clean the sample. Pathogen differentiation from positive blood cultures was described, among other things, by La Scola et al. [98]. The researchers used two protocols to prepare the samples. In the first one, a series of centrifugations were used, and then the pellet was suspended in acetonitrile (AN) and 20% trifluoroacetic acid (TFA) (1:1 *v*/*v*) and incubated for 15 min. In the second protocol, the time of each centrifugation was shortened, and formic acid and ACN (1:1 *v*/*v*) were added to the pellet. After the brief centrifugation, the supernatant was applied to the target plate. The results obtained by the researchers indicate that the first protocol allowed the identification of 94% of Gram-negative bacteria and only 37% of Gram-positive bacteria. Thanks to the use of FA instead of TFA (protocol two), the identification of Gram-positive bacteria increased to 67% and remained high for Gram-negative bacteria (88%). The latest research by Dai et al. describes a fast and simplified protocol to identify microorganisms directly from blood cultures on the basis of the addition of the Triton X-100 reagent and centrifugation [99]. The results of the research show that a high level of identification was achieved for *Enterobacterales* (96.81%), *Enterococcus* (92.31%), non-fermenting *Bacilli* (89.07%), and *Staphylococcus* (88.91%) within 20 min. Researchers confirmed that the identification factor for Gram-positive bacteria is lower than for Gram-negative bacteria.

Identifying microbes directly from biological samples is another challenge. The reason is most likely the difference in the cell wall thickness of these bacteria. Oviaño et al. directly identified bacteria from urine samples using MALDI-TOF MS [100]. They used a Sepsityper kit to prepare the sample. The reliable identification of 91% (503/553) of the samples was obtained by a direct analysis of MALDI-TOF MS urine samples. Identification at the species level was achieved in 88% (487/553) of the samples. Using a direct MALDI-TOF MS analysis, it was possible to identify the main pathogens present in each sample. The mean score for MALDI-TOF MS identification was 2.131. Mohan et al. research show that MALDI-TOF MS’s direct identification can correctly differentiate bacteria in 73.83% of urine samples [101]. Sun et al. developed a new method to diagnose pathogens through MALDI-TOF MS and UF-5000i urine flow cytometers directly from urine samples within 1 h [102]. Ying et al. investigated the possibility of using pathogen enrichment Fc-MBL@Fe_3_O_4_ with MALDI-TOF MS profiling to identify pathogens in samples cultured in liquid. They concluded that Fc-MBL@Fe_3_O_4_ could recognize and trap broad-spectrum microorganisms and could, therefore, be adapted to be combined with the MALDI-TOF MS technique [103].

The MALDI-TOF MS technique can also be successfully applied to identify sporulating bacteria and to analyze compounds that build up bacterial spores. For example, the release of proteins is a general problem in the spore analysis. In order to extract a large mass of analytically useful compounds, bacterial spores are treated with corona plasma discharges (CPD) or subjected to sonication [104]. The results of Ryzhov et al. suggest that the MALDI spectra allow the spores to be characterized as belonging to the *B. cereus* group (*B. anthracis*, *B. cereus* and *B. thuringiensis*) contained peaks that became more visible when the spores were treated with CPD or sonication [105]. In contrast, Afonso et al. used bioactive slides to simplify the analysis of bacterial spores by specific surface absorption and the lysis of spores with strong acids [106]. Horneffer et al. applied wet heat treatment of portions of spore solutions of *B. subtilis*, *Bacillus cereus*, and *B. sporothermodurans* using two techniques. In the first one, deionized water was added to the spore suspension, while the samples were heated in a glycerin bath at 120 °C for 3 and 20 min, respectively, and then immediately cooled. The second technique was to heat the spore solutions in a water bath at 100 °C for 15–30 min. The research results show that both techniques of the wet heat treatment allowed for the release of characteristic proteins from bacterial spores, and thus improved the quality of the obtained protein profiles [28].

#### 3.1.3. Identification Problems

The analysis of higher molecular weight proteins is problematic, because they do not ionize efficiently by MALDI from such a complex mixture as untreated lysate of bacterial cells. It also remains difficult to distinguish other closely related microorganisms such as the *Mycobacterium tuberculosis* complex, including *M. africanum*, *M. caprae*, *M. bovis*, *M. microti*, *M. pinnipedii* and *M. canettii* [107]. Moreover, species with a low index of differences in their ribosomal protein sequences include *Shigella* spp., *E. coli*, certain *Stenotrophomonas maltophilia*, *Propionibacterium acnes* and *Streptococcus pneumoniae*. Members of the *Streptococcus oralis*/*mitis* group may be misidentified by MALDI-TOF MS [63]. Złoch et al. investigated the possibility of MALDI-TOF MS being applied to distinguish closely related salivary streptococci. They proved that the technique could correctly identify streptococcal bacteria by protein and lipid profiling. They also showed that comparable results could be obtained using the FTIR technique, but the interpretation of the breakouts required more time and technical experience [47]. Detecting specific coagulases allows for species differentiation from the *Staphylococcus* genus [108].

Pierce et al. decided to identify *Coxiella burnetii* as a highly infectious microorganism causing Q fever in humans, currently considered a potential bioterrorist agent in the US. Due to the high biological risk, the bacteria were exposed to gamma radiation before the MALDI-TOF MS analysis, which eliminates the viability of *C. burnetii*. The analysis method was validated by predicting unknown samples of *C. burnetii* in an independent test kit with 100% sensitivity and specificity for five of the six strain classes. The supervised pattern recognition via Partial Least Squares-Discriminant Analysis (PLS-DA) was used to confirm the correctness of identification [109].

Jones et al. were the first to describe the use of the Fourier transform mass spectrometry (FTMS) combined with MALDI to analyze bacterial proteins directly from whole cells. It was shown that the accurate MALDI-FTMS mass can be used to characterize specific ribosomal proteins directly from *Escherichia coli* cells. Accurate mass measurements and high-resolution isotope profile data confirm the posttranslational modifications previously proposed based on low-resolution mass measurements. Seven ribosomal proteins were observed from whole *E. coli* cells with errors less than 27 ppm. This was achieved directly from whole cells without fractionation, aggregation, or overexpression of characteristic cellular proteins [110].

#### 3.1.4. ProteinChip Arrays

BioRad introduced ProteinChip Arrays with surfaces that selectively nurture proteins. Therefore, the company renamed MALDI-TOF MS to Surface-Enhanced Laser/Desorption Ionization Time of Flight Mass Spectrometry (SELDI-TOF-MS). The ProteinChip technique is a de novo approach to protein discovery where prior knowledge of specific proteins is not required. The essential elements of the described technology are ProteinChip arrays, ProteinChip reader and dedicated software. ProteinChip arrays are produced using different chemical properties of the surface (Figure 3), and according to Shah et al., three types of matrices, hydrophobic (H50), strong anion exchange (SAX/Q10) or weak cationic (CM10), can provide broad proteome coverage in all microorganisms [111]. Biological samples such as cell lysates, extracts or body fluids are applied to the ProteinChip Array, which allows proteins to bind to the surface based on chromatographic properties or specially designed biological affinity. Unbound molecules are flushed out, and proteins retained on the surface of the template are analyzed and detected using SELDI-TOF MS and the ProteinChip Reader. The obtained MS spectra are compared using differential protein mapping techniques, where the relative expression levels of specific molecular weights are compared using statistical and bioinformatics methods [112].

Research by Rajakarun using the CM10 ProteinChip Array captured the most comprehensive wide range for *S. aureus* isolates [113]. This was also confirmed by the study of Shah et al., who, thanks to the SELDI-TOF MS technique and CM10, correctly differentiated *S. aureus* strains differing in resistance to methicillin [114]. Schmid et al. used a hydrophobic reversed-phase H50 surface to identify *Neisseria gonorrhoeae* causing gonorrhea. Preliminary studies of *N. gonorrhoeae* strains revealed subtle differences in mass spectral profiles, suggesting that SELDI-TOF MS is capable of detecting small differences in the protein expression between strains [115].

### 3.2. Lipidomic 

A lipidome describes the complete lipid profile in a cell, tissue, or the entire body. Lipids are the main functional components of bacterial cells, which play a fundamental role in bacterial metabolism, energy storage and cell signaling. They constitute a barrier between cells and the external environment [116,117]. The cell membrane is the largest lipid reservoir in a bacterial cell. Lipids require modification to perform various functions in the cell and outside it, and the fatty acids themselves differ in the length of the chain and the number of double bonds [118]. The differences were also demonstrated in the lipid profiles of the cell walls of bacteria belonging to the same Gram type [119]. It is essential, apart from lipids present in all bacteria, to characterize the ones that are specific for a given species, showing a high degree of variability, thus enabling the species identification. Gram-negative bacteria are characterized by the presence of a membrane composed of lipopolysaccharide (LPS), which is an amphiphilic endotoxin. LPS is composed of lipid A, an oligosaccharide core and an O-antigen. The lipid A or O-antigen structure differs between the different species of Gram-negative bacteria [120,121,122]. On the other hand, Gram-positive bacteria contain glycolipids, glucolipids and lipoteichoic acid (LTA), which is a typical component of the cell membrane.

Fischer distinguished five types of LTA (I-V) [123]. Type I is the most common characteristic of *S. aureus*, *B. subtilis*, *Enterococcus faecalis*, *L. monocytogenes*, *S. agalactiae* and *Streptococcus pyogenes* [124]. Type II and III were found in *Lactococcus garvieae* and *Clostridium inoculum*. They contained repeating units of glycosylalditol phosphate [125]. In type IV, LTA and WTA (wall teichoic acid) are substituted with choline, mainly in *S. pneumonia* [126]. Type V includes mainly macroamprophiles, such as lipoglycans [127]. Moreover, in mycobacteria, unique long-chain fatty acids—mycolic acids—forming the outer membrane of bacteria were described [128,129,130]. Mycolic acids evince large structural differences, including changes in the chain length (from C_60_ to C_90_), the saturation level, and changes in chemical groups such as ketones and methoxy [131]. In addition, it was demonstrated, for example, that sulfolipid and polyacyltrehalose occurred exclusively in *M. tuberculosis*, while trehalose polifleate occurred in non-tuberculosis mycobacteria [132,133].

The possibility of identifying bacteria based on lipids was first introduced in the 1960s by Abel et al. using gas chromatography (GC) [134]. However, the GC analysis of fatty acids is time-consuming and the sample preparation is labor-intensive, being based on their derivatization to methyl derivatives. Therefore, lipidomics, based on mass spectrometry and combined with other analytical techniques, became a vital tool for the lipid analysis in cells, tissues and even whole organisms [135]. The increasing interest in microbial lipidomics led to the rapid development of lipid analysis techniques using MALDI-TOF MS. Research shows that the lipid analysis by MALDI TOF MS may be a promising tool for detecting antibiotic resistance, for example, in the case of rapidly spreading polymyxin resistance [136]. Commercially available databases used in the MALDI-TOF MS analysis, such as Bruker Biotyper or Vitek, were developed initially to elaborate on the protein profiles of bacteria [47,137,138]. The conducted studies show that lipidomic structures may be as strong or stronger than those based on proteomics [139,140,141,142].

The analysis of lipids or fatty acids requires the selected method of the extraction of these from the cells, such as extraction with organic solvents, division and concentration. The most commonly used methods are the Folch extraction, based on chloroform:methanol (2:1, *v*/*v*) solvents and Blight and Dyer approaches based onchloroform and methanol solvents (1:2, *v*/*v*) with the addition of aqueous salt solution to wash out the polar components [143]. Matyash et al. presented a new extraction protocol developed for profiling complex lipidomes. The method involves the extraction of lipids with methyl tert–butyl ether (MTBE)/methanol, which greatly simplifies sample handling and enables the automated processing of small quantities of biological samples. It was also found that lipid recovery from *E. coli* is the same or better than that obtained by the Folch method [144]. Leung et al., using MALDI-TOF in the negative ion mode, investigated the possibility of identifying ESKAPE pathogens (*Enterococcus faecium*, *S. aureus*, *Klebsiella pneumoniae*, *A. baumannii*, *Pseudomonas aeruginosa* and *Enterobacter* spp.) based on membrane glycolipids, especially glycolipid A. The technique is potentially an alternative to the currently used diagnostics [140]. Scientists used the hot ammonium isobutyrate microextraction protocol developed by El Hamidi et al. [145]. A new method of the rapid lipid extraction from the cell membrane using hot sodium acetate lysis Buffet was presented in the work of Liang et al. This method, combined with the database described above, enables the identification of pathogens from mono- and multi-organism samples in less than one hour [142].

After the extraction, these lipids could be analyzed by MALDI-TOF MS with an appropriate template to allow their ionization and desorption. Walczak-Skierska et al., for the analysis of lactic acid bacteria strains, used the HCCA originally developed for protein analysis [139]. Złoch et al., in addition to HCCA, used the DHB to identify the lipids of salivary streptococci [47]. Angelini et al. used 9-aminoacridine (9-AA), a matrix initially developed to rapidly analyze glycerophospholipids, for the direct lipid analysis of the highly halophilic archaea *Halobacterium salinarum* [146]. Another MALDI matrix, 1,8-bis(dimethylamino) naphthalene (DMAN), was used to identify the lipids of intact Gram-positive *Lactobacillus sanfranciscensis* and *L. plantarum* [147]. Larrouy-Maumusa et al. developed a new method that allows the direct MALDI-TOF MS analysis of lipids on intact microbes [148]. The main advantage of this approach is simple, quick sample preparation that does not require any chemical treatment or purification before the MALDI-TOF MS analysis. The heat-inactivated microorganisms are washed three times in double-distilled water and deposited on a MALDI target plate, and then on a MALDI matrix consisting of a 9:1 mixture of dihydroxybenzoic acid and 2-hydroxy-5-methoxybenzoic acid (super-DHB) dissolved in an a polar solvent system [148,149]. Using this approach, the microbial identification can be completed in less than 10 min for less than 1000 bacteria, making it a useful tool in the clinical laboratory [150]. This method has been hitherto used to differentiate *Mycobacteria*, filamentous fungi and the detection of lipid A in Gram-negative bacteria [151,152,153]. In the case of *Mycobacteria*, this approach provides accurate identification with the sensitivity and specificity of 96.7 and 91.7%, respectively. This method is also quite fast without sophisticated preparation steps. Additionally, studies have reported that using the negative ion mode favors the identification of *Mycobacterium tuberculosis* (Mtb), while the positive ion mode is better for detecting non-tuberculosis mycobacteria (NTM) [150]. Khor et al. presented a new and simple method to detect subspecies-specific lipids used in the *Mycobacterium abscessus* complex (MABS) [154]. The researchers also used the super DHB matrix at a 10 mg/mL concentration but dissolved it in ethanol at 10, 25, 50, 70 and 100%. They also performed the raw mass spectra analysis for 5 McFarland dilutions (5, 10, 20, 30 and 50). They found that combining a matrix called super-DHB with 25% ethanol with a suspension of bacteria in McFarland 20 gave solid and reproducible data, enabling the discrimination of bacteria within strains of the MABS complex. Nevertheless, regardless of the ethanol concentration used and the McFarland dilution, the mass spectra showed two specific peaks for these bacteria.

Cox et al. developed an innovative laser metal oxide ionization (MOLI), which used cerium as a catalyst to convert bacterial lipids into taxonomically available fatty acids. Conversion occurs in situ when applying the lipid extracts to the MALDI target plate spotted with CeO_2_. The CeO_2_-MOLI MS method gave 100% accurate identification at the species and genus level, with only 2% of the incorrect identification at the level of the *Acinetobacter* strain [155]. Using the same technique, the strain level identification was obtained with 94% accuracy among nine different strains of the three *Staphylococcus* species using their fatty acid profiles [156]. Importantly for clinical microbiology, MOLI MS allowed for the correct identification of *Shigella* isolates, a bacterium routinely classified as *E. coli* based on the protein analysis [157]. However, overlapping bacteria and mammalian fatty acids may complicate the direct analysis of patient samples [140].

The possibility of identifying microbial lipids directly from bodily fluids such as blood, urine, and serum, thus omitting the stage of growing microorganisms on agarose plates or in a liquid medium, could constitute a breakthrough in microbiological clinical diagnostics. Leung et al. proved this by adding *S. aureus* or *K. pneumoniae* B6 to blood samples, incubating the bacteria for 6 h and recovering them by differential centrifugation. The lipids from the recovered microorganisms were then extracted and analyzed by MALDI-TOF MS in a negative ion mode. It was then possible to obtain excellent quality signals for 10^4^ CFU bacteria [140].

The challenge in the analysis of bacterial lipids by the MALDI-TOF MS technique is the selection of an appropriate extraction method along with a suitable matrix to obtain the mass spectra of the best possible quality. The MBT Lipid Xtract™ kit (Bruker, Germany) is commercially available, which facilitates efficient sample preparation for microbial lipid analysis, as well as also SimLipid^®^ software (Premier Biosoft, Palo Alto, CA, USA, SimLipid v.6.05 software), compatible with Bruker systems [158,159]. Despite a significant progress in optimizing the sample preparation, a further development of bioinformatics resources is needed to make the lipid analysis user-friendly in clinical diagnostic settings, including building robust and accurate databases. The development of representative lipid databases of comparable size and diversity to the protein mass spectral libraries currently provided by commercial systems will improve the efficiency of microbial identification. Ryu et al. developed and tested a model spectral library for the MALDI-TOF-MS data analysis of bacterial membrane glycolipids, such as lipid A from Gram-negative bacteria and related species from Gram-positive bacteria. Their accomplishment may be important in improving the lipid analysis [141]. A significant drawback of the lipid analysis using the MALDI-TOF MS technique is primarily the background chemical noise arising from the matrix [160]. Another problem arises from the fragmentation in the source of some fragile lipid types. Some of these limitations can be addressed by using alternative matrices to minimize the fragmentation, higher pressure ion sources, and also by using MS/MS to filter out the background chemical noise [161].

### 3.3. Metabolomic

The metabolome is a complete set of small molecules (>1.5 kDa) involved in the metabolism present in cells [162,163,164]. Bacterial metabolism is a highly complex source of bioactive compounds, many of which have significant consequences for human, animal and plant health [165]. The unique bacterial metabolites are analyzed and used for microbial identification, antibiotic resistance development and also as biomarkers for disease detection [166]. The metabolites are classified into primary and secondary metabolites. Primary metabolites are directly involved in the proper development of the body. Secondary metabolites play a significant ecological role and arise during the stationary phase of bacterial growth. The entire metabolites secreted outside the microbial cell are called the exometabolome [167]. The analysis of the exometabolome provides information on the microbial activity under various culture conditions, which, combined with the intracellular metabolic profile, provides a comprehensive overview of microbial metabolism [168].

Microbial metabolomics is becoming more and more widespread in many areas of microbiology and infection research [169,170]. For example, this method proved effective in distinguishing between different strains of *Bacillus cereus* [171], characterizing and differentiating drug susceptibility phenotypes in *Leishmania donovani* [172] to identify volatile metabolites in different *P. aeruginosa* strains [173] and to describe the metabolic adaptations of *P. aeruginosa* strains colonizing various niches in the lung of cystic fibrosis [174]. Kamari et al. conducted tests to detect condensates of volatile organic compounds (VOCs) in three bacteria: *P. aeruginosa*, *A. baumannii*, and *K. pneumoniae*, which may be biomarkers used to identify these bacteria. They found common and characteristic compounds for the given bacterial species [175]. The concept of using microbial VOCs as ‘signature markers’ could provide a faster and non-invasive diagnosis. Finding biomarkers is difficult due to the specificity required in complex matrices. Research by Maurer et al. shows that the VOC sets produced by *Mycobacteria* ssp. change over time and that different strains produce different VOCs [176]. Studies by Moyne et al. found differences in the virulence-related metabolome in several clinical *P. aeruginosa* strains isolated from inert infections in hospitals across Europe [177]. The best known bioactive bacterial metabolites influencing human health include short-chain fatty acids such as propionate, butyrate and acetate, which significantly impact inflammatory bowel disease and colitis [178,179,180].

Laser desorption/ionization mass spectrometry (MALDI-MSI) imaging is a powerful tool for visualizing bacterial metabolites in microbial colonies and biofilms and their interactions. This technique enables a direct visualization of the spatial distribution of metabolic signals by collecting spectra at specific locations in the sample. The active application of IMS with MALDI-TOF for the analysis of microbial samples grown on agar did not begin until the Dorrestein laboratory [181,182]. Advanced software allows for the processing and display of the obtained mass spectra in heat maps, which reveals the location and relative intensity of the analytes. Using MALDI-MSI, Bleich et al. identified *m*/*z* peaks showing spatial distributions superimposed on fluorescence and thus representing putative biofilm-stimulating molecules. This method allowed them to predict and confirm that the thiazolyl antibiotics and thiocillins were metabolites produced by *B. cereus* that stimulated the expression of biofilm genes in *B. subtilis* [183]. The use of MALDI-MSI allows carrying out the rapid identification of metabolic differences between closely related strains. This is confirmed, among other things, by a comparative study of *Lysobacter* strains grown along with *Rhizoctonia solanii*. Comparing monocultures with fungal cultures illustrates how phylogenetically related species can overflow various metabolomic profiles. Moree et al. used MALDI-TOF and MALDI-FT-ICR imaging mass spectrometry (MALDI-MSI) in conjunction with MS/MS networks in their research. This enabled the visualization and identification of metabolites secreted by *P. aeruginosa* and *A. fumigatus* into the culture medium and they observed interactions between organisms at the molecular level [184]. Additionally, mapping the metabolic profiles of *Lysibacter* strains allowed identifying specialized metabolites, which was predicted on the basis of the genome analysis [185]. Moreover, MALDI-MSI enables the observation of complex metabolic pathogen–host interactions in vivo along with the study of other small molecules, including pharmaceutical compounds. An example would be pathogen–pharmaceutical interaction analysis using MS/MS methods to monitor the antibiotic moxifloxacin in the lungs of rabbits infected with *M. tuberculosis*. The obtained results allowed the researchers to conclude that the drug distributor was heterogeneous within the kernels, with relatively lower concentrations in the kernel centers identical to the sites of infectious outbreaks [186]. In the case of MALDI-FT-MS, Jones et al. studied *E. coli* lipids in the low-mass region (*m*/*z* 100–1000) [187]. They identified two main components, phosphatidylethanolamine and triglycerides, commonly found in prokaryotic membranes. The same group described *Saccharomyces cerevisiae* lipid analysis methods with conventional MALDI-FTMS [188].

Chase et al. developed a data acquisition and bioinformatics (IDBac) technique for metabolomic identification that uses the MALDI-TOF MS method to analyze the spectra of proteins and metabolites recorded from single bacterial colonies [189]. It attempts to combine protein testing with the metabolite analysis to distinguish easily between closely related colonies. This technique organizes bacteria into similar phylogenetic groups and allows the metabolic differences of hundreds of isolates to be compared in just a few hours. The IDBac system is free of charge and only requires access to a MALDI-TOF mass spectrometer. Scientists validated the system’s performance by distinguishing the two strains of *B. subtilis* in less than 30 min based on their differing ability to produce the cyclic peptide antibiotics surfactin and plipastatin.

Nguyen et al. adapted the METASPACE cloud software to image the MS metabolite database. Scientists showed that the software used, in conjunction with the relevant specialized metabolite database, could describe specialized microbial metabolites on the basis of agar MS imaging data. This is evidenced by the description of 53 ions representing 32 specialized metabolites validated against the correct taxonomic classification in The Natural Products Atlas [190].

Extremely dangerous secondary metabolites that can lead to acute and chronic poisoning (also fatal) can also cause allergies, fungal infections, respiratory, gastrointestinal and liver diseases. Numerous diseases associated with a weakened immune system are caused by mycotoxins produced by fungi of the genera, among others, *Aspergillus*, *Penicillium* and *Fusarium* [191,192]. Mycotoxins are heat stable and exhibit high levels of bioaccumulation. Hleba et al. investigated the detectability of six different types of mycotoxins: aflatoxin B1, citrinin, deoxynivalenol, zearalenone, T2 toxin, and griseofulvin. The researchers learned that it is possible to detect mycotoxins using a MALDI-TOF Microflex LT mass spectrometer operating in a linear positive ion mode and using an HCCA matrix in a very short time [193]. Very similar conclusions are drawn by the researchers focusing on the identification of mycotoxins produced by *Alternaria* [194] or *Fusarium* [195] fungi.

Mass spectrometry-based metabolomics generates a wide array of data with a very large number of peaks, especially in the case of MSI, where one data set consists of thousands of pixels, each represented by an information-rich mass spectrum. Spectral information is affected by many factors, such as known adduct formation and less-characterized chemical background signals. There is a lack of easy methods to distinguish background chemical signals from actual metabolite signals. [196]. One of the main reasons for the lack of signal identification at the molecular level is the structural diversity (isobars and isomers) and the dynamic range of metabolites. In addition, there is a lack of commercial analytical standards (only a few thousand are available) that are needed for metabolite identification [197].

### 3.4. Genomic

Genomics is a branch of molecular biology that deals with the analysis of the complete genetic material of organisms—the genome. The bacterial genome is a circular DNA molecule called the bacterial chromosome. Additionally, prokaryotes have circular, extra-chromosomal DNA molecules called plasmids containing virulence and antibiotic resistance genes. Whole-genome sequencing has become the gold standard in studying bacterial phylogenetic relationships. The 16S rRNA (rDNA) gene is the most commonly used for bacterial identification. 

The first person to use MALDI-TOF MS to detect DNA was Hurst in 1996. He used 3-hydroxypicolinic acid as a template and detected 108- and 168-basePCR products specific to *Legionella* in the negative ion mode [198]. Two years later, the scientist used the same method to identify genes characteristic of the bacteria *Methylosinus* and *Methylomicrobium*. The measurement of DNA by MALDI-TOF MS is limited as double-stranded nucleic acid is usually not detected due to the acidic nature of the templates. The MALDI resequencing (MALDI-RE) method uses RNA molecules that are more stable during the MALDI ionization. This method is based on the PCR amplification of several ordinal or tandem repeat genes (VNTR). The resulting amplicons are transcribed in vitro; the products are cleaved with specific RNases. The mass spectra of desalinated RNA fragments are measured linearly with a mass range up to 10,000 *m*/*z*. Then, the spectra are compared with the results of the in silico analysis to identify the tested microorganisms. The agreement between the described method and multi-locus sequence typing(MLST) constitutes over 98% [199].

In the early 2000s, von Wintzingerode et al. developed a rapid 16S rRNA (16S rDNA) gene identification approach to the microbial identification, combining the uracil–DNA–glycosylase (UDG)-mediated fragmentation of PCR products with MALDI-TOF MS. The amplified 16S rRNA sequences in the presence of deoxyuridine triphosphate (dUTP) instead of thymidine triphosphate (dTTP) were immobilized on streptavidin-coated solid supports [200]. This enabled the selective production of sense and antisense matrices. The single-stranded PCR products were then treated with uracil-DNA glycosylase to generate T-specific basic sites. The amplicon fragmentation by the base treatment was also performed. MALDI-TOF MS was then used to analyze the resulting fragment patterns. The study distinguished between *Bordetella* species and closely related *Alcaligenes* and *Achromobacter.*

On the basis of the in silico pattern database of proprietary sequences and the 16S rDNA sequence database, 24 mycobacterial isolates were correctly identified. Repeated experiments showed high reproducibility. The platform is not limited to identifying 16S rDNA. Still, it can be extended to other genotypic markers, e.g., *gyrB* sequence polymorphism analysis for differentiation of the *M. tuberculosis* complex, multi-drug resistance regions or multilocus sequence typing, further broadening its application [201]. Cuénod et al. identified clinically essential and now often misdiagnosed *Klebsiella* spp. They used the whole-genome sequencing (WGS), the comparative genomic analysis and the in silico protein mass prediction of ribosomal subunits from WGS data. A diverse selection of bacterial isolates (n = 50) representing at least eight *K. pneumoniae* isolates sequenced throughout the genome was used to validate the detection of the predicted marker masses in the MALDI-TOF mass spectra. Based on a systematic comparison of WGS and in silico ribosomal mass prediction, researchers proposed a MALDI-TOF MS analysis to discriminate between eight *Klebsiella* species [202].

Dunne et al. developed a sequence-typing method MLST by mass spectrometry, making it possible to compare the peak patterns of the cleavage fragments of specific PCR amplicons with the reference sequence assigned to isolates sequence types. Such an analysis may increase the chance of identifying subgroup-specific biomarker peaks. MLST is used for the global surveillance of bacterial pathogens. Through MLST and MALDI-TOF MS, the same researchers discovered seven new alleles and 30 previously unreported sequence types of *S. pneumoniae* isolates [203]. Ha et al. used The Korean TrueBac ID system to identify the bacteria on the basis of the entire genome of the bacteria. They proved that the method used could differentiate pathogens in opposition to the standard MALDI-TOF MS method and 16S rRNA sequencing [204].

Sequenom InC. launched the MassARRAY system combining MALDI-TOF MS with endpoint PCR. Currently, it is still not used in the clinical diagnosis of microbiology. The system enables the identification of microorganisms based on the target DNA transcription and base-specific RNA cleavage. The usefulness of this technique in the differentiation of microorganisms was confirmed by many researchers [205,206,207,208].

## 4. Antibiotic Resistance

The effectiveness of and easy access to antibiotics has led to their overuse, which, in turn, has led to the development of drug resistance among microorganisms [209,210]. The consequences of the emergence of drug resistance were the discovery, development and implementation of new β-lactam antibiotics, which started in 1950, the period known as the “golden age of antibiotic discovered” [211,212]. Soon, the first cases of methicillin-resistant *S. aureus* (MRSA) were discovered [212]. As a result, the effectiveness of antibiotics in treating bacterial infections has significantly diminished in the recent years. The consequence of this is the development of new antibiotics. Nevertheless, the effectiveness of antibacterial drugs in treating bacterial infections has decreased in the recent years. The continuous selection pressure of various drugs has led to the emergence of bacteria with additional resistance mechanisms, resulting in the emergence of multi-drug resistant bacteria (MDR), extensively drug-resistant bacteria (XDR), and pan-drug resistant bacteria (PDR), called superbugs [213].

Serious infections include nosocomial infections due to multidrug-resistant strains of bacteria such as *A. baumanii*, *C. difficile*, *Enterobacter* spp., vancomycin-resistant *enterococci* (VRE), *E. coli*, *Haemophilus influenzae*, *K. pneumoniae*, *P. aeruginosa*, *Staphylococcus epidermidis*, *S. pneumoniae*, extensively drug-resistant *M. tuberculosis* (XDR-TB) or MRSA, considered to be the most common superbug [213,214,215,216]. 

Furthermore, with the advent of incurable strains of Enterobacteriaceae resistant to carbapenems, humanity was on the threshold of the post-antibiotic era [217]. Carbapenem resistance is one of the most troublesome for antibiotic resistance, as infections with carbapenem-resistant bacteria have a 48% mortality rate [218].

The “gold standard” used to assess the susceptibility of microorganisms to the antibiotics used is the disc diffusion method introduced by Bauer and Kirby in 1956, using the phenomenon of the formation of a concentration gradient in the substrate during the diffusion of the active substance from the antibiogram disc [219]. A significant achievement in the routine analysis of widespread antibiotic resistance in bacteria was the development of epsilometer testing (E-test). Plastic E-test strips are coated with predefined concentrations of antibiotics, and the appropriate minimal inhibitory concentration (MIC) ranges are marked on the surface of the strip [220]. The identification of antibiotic resistance can also be made by searching for the homology of DNA sequences against a database. For this purpose, several reference databases were designed, including the Antibiotic Resistance Gene Database (ARDB) [221], Strucured Antibiotic Resistance Gene Database (SARG) [222], Comprehensive Antibiotic Resistance Database (CARD) [223] and ResFinder [224].

The first successful use of MALDI-TOF MS to detect antibiotic resistance was the observation of β-lactam ring hydrolysis after antibiotic exposure to β-lactamases produced by aerobic and anaerobic bacteria. The evidence of positive results was demonstrated on mass spectra where signals characteristic of the drug used and its hydrolysis products were visible. Other LDI techniques, such as nanotechnology-assisted laser desorption/ionization time-of-flight mass spectrometry (NALDI-TOF MS) and SELDI-TOF MS, were also used to identify antibiotic resistance. Similar to identifying microorganisms, resistance to antibiotics can occur through proteomics, lipidomics, metabolomics, and genomics.

### 4.1. Proteomics

In *Staphylococcus,* there are methicillin-resistant (MRSA) and methicillin-sensitive (MSSA) strains [225]. This makes it difficult to classify these strains into two different groups. Partial success was achieved by developing a method involving the production of a phenol-soluble protein toxin (PSM-mec) by a subset of MRSA strains that is detectable by MALDI-TOF MS at the peak of 2415 ± 2.00 *m*/*z*. The “MBT Subtype Module” software (Bruker Daltonics GmbH, Bremen, Germany, flexAnalysis software version 3.4) was developed to detect PSM-mec in the mass spectrum of *S. aureus* isolates, providing the indirect evidence of methicillin resistance [226,227]. However, the use of PSM in identifying MRSA is not very reliable. This method has high specificity, but low sensitivity and is therefore no longer used as commercially available software [228,229] In addition, there are other examples of the subtype module. One of them is the use of MALDI-TOF MS and the CarbaNP test for the rapid identification of *Bacteroides fragilis* strains with the cfiA gene, which is responsible for developing resistance to carbapenems [228].

Researchers recently reported a method to determine bacterial sensitivity using the direct-on-target microdroplet growth assay (DOT-MGA), which is straightforward, practical, and quick to perform. For the DOT-MGA analysis, the bacteria are incubated with and without the indicator antibiotic in the microdroplets nutrient broth directly on the MALDI-TOF MS targets. The evaporation and drying out of places is solved by storing the plate in plastic containers with water at its bottom. By assessing the growth in the presence of various antibiotics, it is possible to determine the sensitivity of the isolate and analyze the potential mechanisms of drug resistance. Nix et al. investigated the possibility of using DOT-MGA to rapidly detect MRSA in patients with positive blood culture bottles using cefotaxitin. The researchers calculated that the optimal size of the microdroplets is 6 µL [229]. The authors analyzed three sample preparation methods: continuous broth dilution, lysis/centrifugation, and differential centrifugation. The results showed that lysis/centrifugation and 4-h incubation led to the best reliability, sensitivity and specificity. In the MRSA analysis, an additional step was to destroy the cell membrane by adding formic acid before adding the matrix. This step could be omitted when performing DOG-MGA on Gram-negative bacteria. Idelevich et al. determined carbapenemase resistance in *K. pneumoniae, E. cloacae, E. aerogenes, P. mirabilis*, and *K. aerogenes*. Additionally, they examined the fourth pretreatment method, filtration/dilution, but observed that lysis/centrifugation and a 4 h incubation of bacteria gave the best results [230].

Another test based on MALDI-TOF MS is the resistance test that detects isotope-labeled, stable (non-radioactive) amino acids that are incorporated into newly synthesized bacterial proteins (MBT RESIST) [231]. In this approach, the deficiency of the corresponding amino acids in the culture medium is supplemented with either radio-labeled amino acids or labeled amino acids in combination with the antibiotic to be tested. The culture media labeled with specific isotopes is the main limitation of this approaches [232]. Microorganisms are grown simultaneously on two different media, one containing the 12C isotope and the other with 13C as the carbon component. The susceptibility/resistance of bacteria is determined by the amount of radio-labeled amino acids incorporated into the newly synthesized proteins. The growth of resistant bacteria is observed in the presence of an antibiotic containing 13C in its polypeptides. This shifts the signals to higher *m*/*z* values in the mass spectrum. The MBT RESIST approach was used to examine antibiotic resistance in MRSA strains using oxacillin and cefoxitin and to detect ciprofloxacin, meropenem and tobramycin resistance in *P. aeruginosa* after 3 h of incubation [231]. The method’s difficulty is that the masses of proteins represented by the tested signals must be previously known and conserved for all species strains [228].

Antibiotic resistance can also be determined using semi-quantitative mass spectrometry using the MBT ASTRA approach, which can be used for all antibiotics and microbial species classes. The technique consists of calculating and comparing the areas under the curves (AUC) of the spectra of bacteria exposed or not to the antibiotic [229]. If the microbial strain is susceptible, the AUC of the bacterial suspension with the antibiotic will be lower than it would be without it. In contrast, the AUC with or without the antibiotic will be comparable for the resistant strain. Ceyssens et al., using MBT ASTRA, successfully assessed the sensitivity of *M. tuberculosis* strains to rifampin, isoniazid, linezolid and ethambutol, as well as non-tuberculous mycobacteria to rifampin, isoniazid, linezolid and ethambutol [228]. The main disadvantage this approach is complicated and a multistep procedure, with has limited its common application in routine analysis [232]

### 4.2. Lipidomic

Changes in the lipid composition of bacteria are also associated with developing drug resistance. A common mechanism of resistance is the modification of the lipopolysaccharide (LPS), which, among others, includes the modification of the A-lipid regions. In addition, the use of antibiotics enables the release of lipopolysaccharide (LPS), which is an important cause of the development of septic shock in patients treated for severe infections caused by Gram-negative bacteria. It is generally accepted that LPS from the outer membranes of Gram-negative bacteria is responsible for many of the clinical symptoms of sepsis [233].

The resistance of ESKAPE pathogens to colistin is given by the mcr-1 gene encoding phosphoethanolamine transferase (PE). Its expression leads to the modification of PE lipid A, reducing its total negative charge. The MALDI-TOF MS analysis of three ESKAPE clinical pathogens (*K. pneumoniae*, *A. baumannii* and *P. aeruginosa*) with *mcr-1* by Liu et al. demonstrated PE to lipid A and revealed that even strains showing slight decreases in sensitivity had a modification of the PE of lipid A [234]. These results indicate that the MALDI-TOF MS technology may be a valuable tool in monitoring the spread of *mcr-1* among pathogens. Dortet et al. used the MALDxin test based on the MALDI TOF-MS technique to detect *A. baumanii* resistance to colistin. The test accurately detected all colin-resistant bacterial isolates within 15 min with limited sample preparation before the MALDI TOF-MS analysis. Standard methods require 24–48 h to obtain a result [151]. The research by Lopalco et al. in 2017 identified unique acid glycerophospholipids: cardiolipin, and monolisocardiolipin in *A. baumannii* using MALDI-TOF MS thin layer chromatography (TLC). The knowledge of these compounds allows for determining the resistance of bacteria to environmental factors and antibiotics [235]. The rapid detection of colistin resistance based on lipid A modification is also possible thanks to the new MBT LipidArt software (Bruker Daltonics GmbH, Bremen, Germany) using MALDI Biotyper^®^ sirius System analysis in negative ion mode.

Bisignano et al. showed the potential correlation between *S. aureus* lipid profile, the site of infection, antibiotic resistance, and cell surface hydrophobicity [236]. Researchers showed that bacterial lipid profiles differed both qualitatively and quantitatively between different strains of *S. aureus*, and this change affected both antibiotic resistance and cell surface hydrophobicity.

### 4.3. Metabolomic

MBT-ASTRA is a rapid antibiotic resistance detection method based on the MALDI-TOF MS software tool AUC [237]. Another test, the MBT-STAR-BL assay, is already a widely studied functional assay analyzing the bacterially induced hydrolysis of β-lactam antibiotics [238,239]. Hydrolysis is monitored by observing specific mass shifts, which in most cases are detectable after an incubation time of 30–180 min. The suitability of the MBT STAR-Cepha and MBT STAR-Carba tests for detecting bacteria producing extended action b-lactamases (ESBL) and carbapenemases was also assessed. The authors compared the investigated techniques with covert methods such as microdilution or PCR amplification. Studies show that the MBT STAR-Cepha kit effectively distinguished between resistant strains of third-generation cephalosporin-sensitive phenotypes. In addition, the MBT STAR-Carba kit accurately detected antimicrobial resistance by carbapenemase producers. The obtained results suggest that the target bacterial strains, antimicrobial susceptibility phenotypes and resistance genes were necessary for utilizing MBT STAR-Cepha and MBT STAR-Carba kits based on MALDI-TOF MS in routine bacterial diagnosis [240,241].

### 4.4. Genomic

Currently, the most common use of MALDI-TOF MS in genomic research is for genotyping and detecting single nucleotide polymorphisms responsible, among other things, for antibiotic resistance. SNPs are DNA sequence variations that occur when a single nucleotide in a genome sequence is replaced with another [242]. For discovering new SNPs and bacterial fingerprinting, re-sequencing methods may be helpful, which involve sequencing part of an individual’s genome to detect sequence differences between the individual and the standard genome of the species [243,244]. One proposed MALDI-TOF MS-based approach for detecting *K. pneumoniae* resistance to carbapenems is based on identifying plasmids carrying *blaKPC* carbapenemase genes [245,246].

The MassARRAY system involves adding SNP sequence-specific extension primers to the amplified PCR product for a one-base extension at the SNP site. The prepared analytes are then co-crystallized with the chip array and analyzed by MALDI-TOF MS. SNP genotyping on the MassARRAY system combines multiplexed primer extension chemistry with highly sensitive mass spectrometry. This combination provides the precise, rapid, cost-effective analysis of hundreds of genotypes daily. MassARRAY technology allows the analysis of SNP combinations in 96- or 384-well plates. Furthermore, it is possible to analyze at least 40 SNPs per well [247,248]. Si et al. proved that the MassARRAY system based on the MALDI-TOF MS technique detected 60 copies of Mtb gene mutations associated with the emergence of resistance to rifampicin and isoniazid, streptomycin, quinolone or aminoglycosides [249].

Pu et al. used whole genome re-sequencing to obtain differences in genomic levels between *A. baumannii* strains. Diversity was determined by multi-locus sequence typing, and the genetic relationship between ten strains and others was examined by phylogenetic analysis. They conducted a comparative analysis focused on resistance genes related to insertions and deletions and single nucleotide polymorphisms (SNPs) to identify the primary mechanism of *A. baumanii* resistance [250]. Another example is the study by Suzuki et al., who performed genome re-sequencing analyses for each drug-resistant *E. coli* strain tested to identify fixed mutations and changes in gene expression. Moreover, they looked at how acquiring resistance to one drug alters the resistance and susceptibility to other drugs. By integrating this data and using a simple mathematical model, scientists demonstrated how to quantify drug resistance based on the expression levels of a small number of genes [251].

Ikryannikova et al. presented an approach using the MALDI-TOF MS-based micro-sequencing reaction to detect the SNPs responsible for extending the substrate specificity of β-lactamases towards oxyimino-cephalosporins in *E. coli* and *K. pneumoniae*. In this work, the described approach was mapped to the analysis of polymorphisms in three codons of the bla_TEM_ gene. The MALDI TOF MS-based mini-sequencing assay can detect and differentiate all key mutations conferring β-lactamase activity from the wild-type sequence. Different mutations at the same site are distinguishable due to the apparent differences in mass peaks. The main advantage of the developed test is its high level of reliability: all polymorphisms identified by the mini sequencing technique were confirmed by direct DNA sequencing results [252].

## 5. Biofilm and Development of Antibiotic Resistance

Biofilms constitute a protective barrier for pathogens, enabling them to survive in stressful environmental conditions. Research into the development and control of biofilms using new techniques is crucial in medicine and environmental research. Biofilm plays a pivotal role in surviving external threats and toxic materials, including antimicrobial drugs [253]. Biofilm formation occurs in four basic steps: reversible adhesion, irreversible adhesion, biofilm maturation and cell dispersion. At the stage of biofilm maturation, the bacteria begin to secrete extracellular polymeric substances (EPS), which constitute up to 90% of the mature biofilm structure [254]. EPS comprises polysaccharides, proteins, extracellular DNA (eDNA), and lipids [255].

Gram-positive and Gram-negative bacteria can form biofilm on medical devices, implants, and surgical wounds or teeth. It is estimated that *S. aureus* and *S. epidermidis* cause about 40–50% of heart valve prosthesis infections, 50–70% of catheter biofilm infections and 87% of bloodstream infections [256]. Two-thirds of infections associated with implanted devices are caused by staphylococcal species such as *S. aureus* and coagulase-negative staphylococci [257,258]. *P. aeruginosa* quickly adapts to harsh conditions and antibiotics and is widely used as an in vitro model to study biofilm formation [259].

Resistance to antibiotics in the bacterial biofilm occurs as a result of the slow or incomplete penetration of the drug by the polymer matrix, the interaction of the drug with this matrix, a result of which being that the antibiotic loses its properties, the presence of enzymes such as β-lactamases, genetic changes on target cells or hiding target sites, extrusion antibiotics using efflux pumps [260], and the company of an outer membrane structure such as that of Gram-negative bacteria [261]. It was shown that mycolic acids in *Mycobacterium smegmatis* EPS may be associated with higher resistance to antibiotics [262]. Adibi et al. showed the biofilm production was higher in MDR *P. aeruginosa* strains than in strains without MDR [263]. The presence of multi-drug resistance in *A. baumanii* strains also increased the biofilm production, as confirmed by the studies by Amin et al. [264]. Manandhar et al. linked the biofilm production in *S. aureus* to methicillin resistance [265].

On the other hand, many studies fail to prove a direct relationship between the MDR phenotype and the increased bacterial biofilm production [266,267]. Caputo et al. showed that MALDI-TOF MS could be used to differentiate quickly and accurately clinical *S. epidermidis* isolates as biofilm producers. The researchers identified clinical strains derived from suture wires, and their protein profiles were compared with those obtained from two ATCC reference strains (biofilm producer and non-producer). Using the MALDI method they identified eighteen isolates as *S. epidermidis* by matching sixteen profiles to the biofilm producer and two to the non-producer, supporting the crystal violet test results [268].

The use of MALDI-TOF MS also allows for predicting the influence of substances secreted by one bacterium on the impact on the physiology of neighboring microorganisms, including the activation or inhibition of the expression of the biofilm genes. Bleich et al. used MALDI-IMS to identify thiocillins, antibiotics of the thiazolyl peptide group. These compounds produced by *B. cereus* induce the biofilm matrix produced by *B. subtilis*. Researchers found that thiocillin increased the *B. subtilis* cell population. An important observation is that the mutation eliminating the antibacterial activity of thiocillin did not affect the ability to induce the expression of the biofilm formation gene [183].

Li et al. described a one-step, spray application of a 2,5-dihydroxybenzoic acid solution for the direct imaging of desorption/laser ionization of hydrated *B. subtilis* biofilms on agar supported by a matrix. An optimized airbrush and an automatic home sprayer showed the region-specific distributions of signal metabolites and cannibalistic factors from *B. subtilis* cells grown on a biofilm-promoting medium. This approach provides a uniform, relatively dry coating on the hydrated samples, improves the point-to-point signal reproducibility compared to a screened matrix, and is easily adapted to imaging a range of agar-based biofilms [269]. Pauter et al., using the same biofilm imaging technique, showed changes in the molecular profiles of *Bacillus tequilensis* before and after the antibiotic therapy, leading to the proposed antibiotic resistance mechanisms. The researchers showed that the matrix-assisted laser desorption mass spectrometry could be used along with the UniProt database as a complementary technique to capillary electrophoresis (CE) to study differences in the molecular profile of *B. tequilensis* after the antibiotic treatment [270]. Additionally, Si et al. studied molecular heterogeneity in biofilms of *B. subtilis* colonies using MALDI-TOF MS. In this study, they combined the MALDI and fluorescence methods, which allowed detecting distinct populations of cells in a biofilm [271]. Pereira et al. evaluated the MALDI-TOF mass spectrometry to analyze the molecular profile of *P. aeruginosa* biofilms grown on glass and plastic surfaces at different stages of the biofilm development. The results of molecular studies show that MALDI-based profiling cannot distinguish between the various stages of the biofilm development, but this can be observed when biofilm cells are released in the dispersion phase that first occurred on the polypropylene surface [272].

In 2019, De Carolis et al. published a paper presenting a newly developed BIOF-HILO test based on MALDI-TOF MS in conjunction with analyzing protein profiles with a complex correlation index (CCI). Researchers proved that this enabled the rapid (i.e., 3-h) identification of *Candida parapsilosis* isolates with the high or low biofilm formation capacity [273]. However, as no reference *C. parapsilosis* mass spectral databases were used in this test, more research is required before the developed method can be used in the routine patient diagnosis.

Based on the presented research, it can be concluded that MALDI profiling may become a promising technique for a clinical diagnosis and the prediction of the development of biofilm formation.

## 6. Conclusions

Solving the crisis of increasing antibiotic resistance requires discovering compounds with new mechanisms of action and searching for new, more accurate and faster methods of identifying bacteria and their antibiotic resistance. In addition, the availability of updated epidemiological data on antimicrobial resistance in common bacterial pathogens will be helpful in making decisions concerning treatment strategies and developing an effective hospital antimicrobial management program. Combining the MALDI TOF-MS technique with a multiomic approach can bring excellent results in accurately and comprehensively identifying microorganisms and resistance to antibacterial drugs. Currently, the priority is to shorten the incubation time, sample preparation and analyzes, which will significantly speed up the diagnosis of clinical patients. As the MALDI-TOF technology advances and develops beyond the pattern recognition of unfractionated cell lysates and/or intact cells, it will become important and necessary to identify the specific protein ions whose amino acid sequence is unique to the microorganism under study. The identification of protein toxins, virulence factors, and antibiotic resistance mechanisms becomes particularly important. Genomics, proteomics, and metabolomics are necessary to obtain information that is not available from the sequenced genome. Additionally, MALDI-TOF MS techniques have the potential to significantly advance the identification of undigested proteins.

## Figures and Tables

**Figure 1 ijms-23-09601-f001:**
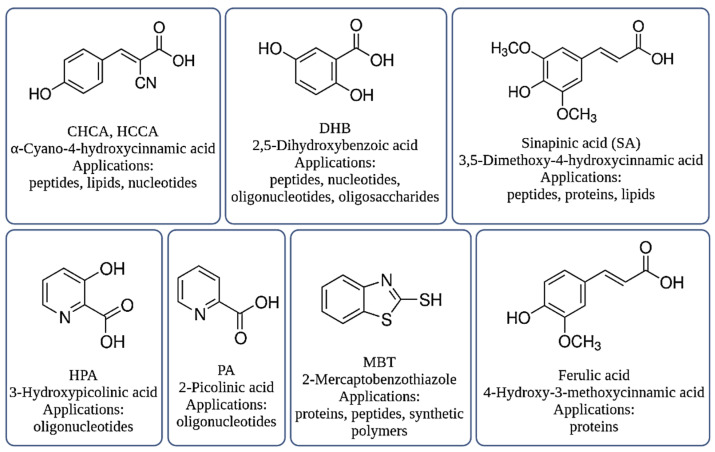
Examples of matrices used in the MALDI-TOF MS technique.

**Figure 2 ijms-23-09601-f002:**
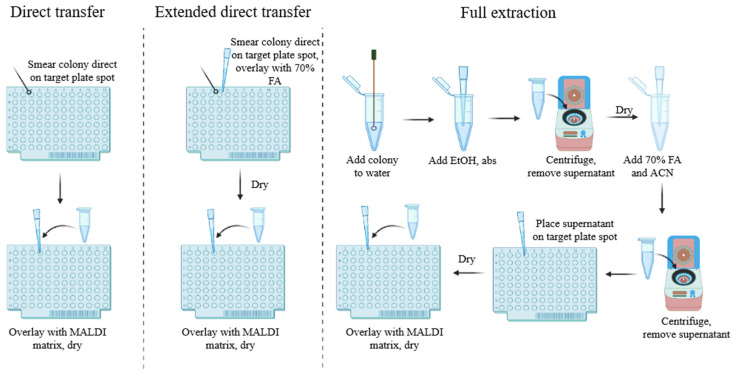
Sample preparation procedures by direct transfer, extended direct transfer, and full extraction using the ethanol/formic acid method.

**Figure 3 ijms-23-09601-f003:**
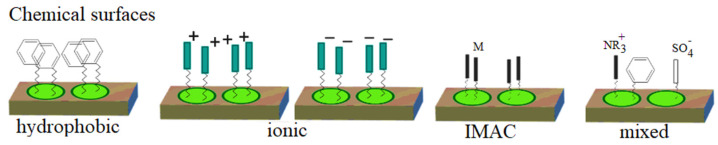
Examples of chemical surfaces in SELDI ProteinChip arrays.

## Data Availability

Not applicable.

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
