# Peer review of "“Omic” Approaches to Bacteria and Antibiotic Resistance Identification"

_ijms, 2022, doi:10.3390/ijms23179601_

Round 1

Reviewer 1 Report

It is very interesting and consistent review on modern approaches to bacteria and antibiotic resistance identification based on a numer of „omic” techniques. Authors present a critical state of art of bacteria identification based on proteomic, lipidomic, metabolomic and genomic approaches.

As a well practised analysts stress an important issue of sample preparation and  antibiotic resistance, followed by biofilm and development of antibiotic resistance.

This manuscript  should be accepted as it is with very minor correction of the title in chapter 6.

I suggest a title „Conclusions”, but definitly not „conclusion”.

Author Response

Thank You very much for your critical review. It was very useful in the correction of our manuscript. Identification of weak points throughout the text has helped us to increase the value of our paper. All comments and changes suggested by Reviewers have been incorporated into the manuscript. Once again, Thank You very much for your help.

Reviewer 2 Report

The given article covers a very interesting and complex topic of using different MALDI-TOF MS methods for identifying clinically important bacteria including detecting their antibiotic resistance. The positive points of this article are its range and good way to explain single subparts of this topic.

The points to be corrected in the minor revisions are as follows.

From the scientific points of view, I have the following comments or questions.

1)   L. 121-123: The main disadvantage of using FA as a matrix, however, is the lack of the automatic acquisition of the 122 mass spectra.

Could you explain why the automatic acquisition is not possible only for FA?  According to my experience the possibility of the automatic acquisition depends only on the sample quality (as the quality to obtain intensive peaks easily), not on the used matrix.

2)   L. 133: … using the MALDI-TOF MS online.

What does it mean online in this context? Would not it be in silico correctly?

3)   L. 164-165 Unfortunately, these results are difficult to compare due to the strong structural differences between bacterial spores and their vegetative cells or fungal spores.

I do not see the meaning of it as the text before is about identifying the bacterial spores, not vegetative cells. I think this sentence should be reformulated or other information about this topic should be added.

4)   L. 179-180 By using different MAL-179 DI-TOF MS modes, a greater variety of results can be obtained from a single sample 180 analysis [37].

Is there really only one study on this topic?

5)   L. 264-266 ...where the value .2 means "high confidence identification", .1.7 and <2 means "low confidence identification", and a score <1.7 is interpreted as "no reliable identification".

This statement is not correct, in Biotyper there are three limits for score values: more than 2.3, between 2.0 to 2.3 (except), between 1.7 to 2.0 (except) with the confidence statements for the probability of species and genus identification. Besides this the consistency of ten best result is another parameter to evaluate the identification. Please correct this statement.

6)   L. 280-282 Hyeon Park et al. compared the identification performance of the recently developed 280 Autof ms1000 (Autobio, China) with that of the Bruker Biotyper (Bruker Daltonics, Germany).

L. 297-299 Korean 297 scientists developed a new MALDI-TOF MS ASTA MicroIDSys system (ASTA, Suwon, Korea).

Could you please specify the principle of the identification in these systems? Is it like Bruker or bioMérieux principles?

7)   L. 289  …using the direct transfer with the additional ethanol treatment.

Is this correct as ethanol treatment is used only in the extraction method?

8)   L. 333-334 Direct sample transfer gives the best identification results for rod-shaped Gram-negative bacteria [78].

Are there some groups of bacteria or other microorganisms for which extended transfer or extraction methods are recommended dominantly?

9)   L. 370-371 Oviaño et al. directly identified bacteria from urine samples using MALDI-TOF MS [86].

    Could you please specify the approach to identify single bacterial (or microbial) species, if they are mixed in the sample?

10) L. 432 Chapter on ProteinChip Arrays.

Could you explain more in details this system?

11) L. 533 with the sensitivity and specificity of 96.7% (Mycobacterium tuberculosis) and 91.7% of (non-tuberculous mycobacterial species).

Are both sensitivity and specify the same in both cases?

12) In the Chapter 2 MALDI-TOF MS principle is described in detail. I miss the same description for other methods or MALDI-TOF MS modifications as e.g. MALDI-FT-ICR, MALDI-FTMS, MALDI-TOF Microflex.

13) It is under question whether Table 1 (l. 272) is really necessary in this article and what is this information value of it. The given information could be mentioned within the appropriate paragraph. Instead of this table if appropriate the table summarizing and comparing different approaches to use MALDI-TOF MS to identify ATB resistance might be formed.

14) L. 884 4.4 Genomic chapter. I miss the information of using MALDI-TOF MS for genotyping (Vogel et al., 2009) or detection of SNP responsible for the antibiotic resistance (Griffin and Smith, 2000;  Sauer, 2006; Ikryannikova et al., 2008). Please involve also this information in your review.

Vogel N., Schiebel K., Humeny A. (2009): Technologies in the Whole-Genome Age: MALDI-TOF-Based Genotyping. Transfusion Medicine, 36, 253-262.

Griffin T.J., Smith L.M. (2000): Single-nucleotide polymorphism analysis by MALDI-TOF mass spectrometry. Trends in Biotechnology, 18, 77-84.

Sauer S. (2006): Typing of single nucleotide polymorphisms by MALDI mass spectrometry: Principles and diagnostic applications. Clinica Chimica Acta, 363, 95-105.

Ikryannikova L.N., Shitikov E.A., Zhivankova D.G., Ilina E.N., Edelstein M.V., Govorun V.M. (2008): A MALDI TOF MS-based minisequencing method for rapid detection of TEM-type extended-spectrum beta-lactamases in clinical strains of Enterobacteriaceae. Journal of Microbiological Methods, 75, 385-391.

The usage of English requires a minor revision in some stylistic parts. There are used some incorrect structures (allow for, allows the testing – l. 105, using “and” as “by how” – l. 87, he used instead of they used– l. 237) or some sentences are difficult to understand (l. 14-15, 275).

The formal side needs also to be improved in some cases as there are  some typing error (l. 440, 478, 510, 592, 87-89: a repeated sentence), missing gaps after final dots (e.g. l. 57, 108, 229, 645), redundant gaps within sentences (e.g. l. 135, 218,  concatenated words (e.g. l. 149, 228, 256, 385, 407, 602, 604) or missing letters (l. 41, 315, 472, 478) or missing italics (l. 34, 240, 426).

Author Response

The given article covers a very interesting and complex topic of using different MALDI-TOF MS methods for identifying clinically important bacteria including detecting their antibiotic resistance. The positive points of this article are its range and good way to explain single subparts of this topic.

The points to be corrected in the minor revisions are as follows.

From the scientific points of view, I have the following comments or questions.

1)   L. 121-123: The main disadvantage of using FA as a matrix, however, is the lack of the automatic acquisition of the 122 mass spectra.

Could you explain why the automatic acquisition is not possible only for FA?  According to my experience the possibility of the automatic acquisition depends only on the sample quality (as the quality to obtain intensive peaks easily), not on the used matrix.

RE: Thank You a lot for Your remarks. The FA matrix does not allow for automatic acquisition of mass spectra because it gives a heterogeneous layer of matrix crystals. This information has been added in the text.

2)   L. 133: … using the MALDI-TOF MS online.

What does it mean online in this context? Would not it be in silico correctly?

 RE:Thank You for remarks. Yes, the “online” has been replaced by in silico.

3)   L. 164-165 Unfortunately, these results are difficult to compare due to the strong structural differences between bacterial spores and their vegetative cells or fungal spores.

I do not see the meaning of it as the text before is about identifying the bacterial spores, not vegetative cells. I think this sentence should be reformulated or other information about this topic should be added.

RE:This sentence was removed because, upon careful reflection, it was inconsistent with the entire paragraph.

4)   L. 179-180 By using different MAL-179 DI-TOF MS modes, a greater variety of results can be obtained from a single sample 180 analysis [37].

Is there really only one study on this topic?

RE: There is much more research on this subject. More references has been added to manuscript

5)   L. 264-266 ...where the value .2 means "high confidence identification", .1.7 and <2 means "low confidence identification", and a score <1.7 is interpreted as "no reliable identification".

This statement is not correct, in Biotyper there are three limits for score values: more than 2.3, between 2.0 to 2.3 (except), between 1.7 to 2.0 (except) with the confidence statements for the probability of species and genus identification. Besides this the consistency of ten best result is another parameter to evaluate the identification. Please correct this statement.

RE:Thank You a lot for your remarks. The sentence has been corrected.

6)   L. 280-282 Hyeon Park et al. compared the identification performance of the recently developed 280 Autof ms1000 (Autobio, China) with that of the Bruker Biotyper (Bruker Daltonics, Germany).

  1. 297-299 Korean 297 scientists developed a new MALDI-TOF MS ASTA MicroIDSys system (ASTA, Suwon, Korea).

Could you please specify the principle of the identification in these systems? Is it like Bruker or bioMérieux principles?

 RE:Thank You for remarks. The missing information has been added to manuscript:Autof MS 1000 ASTA andMicroIDSyssystem, analogous toMALDI BioTyper the database is based on an isolate-specific references approach, while forbioMérieux principles (e.g. Vitek MS), it is based on a taxonomical group-specific principles[1,2]”

[1] https://www.frontiersin.org/articles/10.3389/fcimb.2021.628828/full

[2] https://www.annlabmed.org/journal/view.html?doi=10.3343/alm.2017.37.6.531

7)   L. 289  …using the direct transfer with the additional ethanol treatment.

Is this correct as ethanol treatment is used only in the extraction method?

RE:The described procedure is correct. The researchers applied the bacteria to the MALDI-TOF MS endplate, skipping the extraction step. After drying, they applied ethanol to fix and deactivate the tested microorganisms. This information has been added in the text.

8)   L. 333-334 Direct sample transfer gives the best identification results for rod-shaped Gram-negative bacteria [78].

Are there some groups of bacteria or other microorganisms for which extended transfer or extraction methods are recommended dominantly?

RE:Thank You for remarks. Direct transfer or extended direct transfer will definitely be the better method for Gram-negative and non-endospore producing bacteria.A thick cell wall then makes protein analysis difficult.Direct transfer methods are much faster and allow you to perform more analyzes in less time.Extraction methods, due to higher protein recovery, are more preferable for MALDI detection ofGram-positive bacteria (in particular sporulating bacteria).Missing information has been added to revised version of manuscript.

9)   L. 370-371 Oviañoet al. directly identified bacteria from urine samples using MALDI-TOF MS [86].

    Could you please specify the approach to identify single bacterial (or microbial) species, if they are mixed in the sample?

RE: Direct identification of urine samples in the study by Oviaño et al was based on the identification of the major pathogen in the sample. In addition, the researchers compared the results obtained with the results obtained from the identification of microbes in the culture of the same urine samples. In the study of mixed cultures, it is also possible to use the Bruker Biotyper® mixed culture algorithm, which enables the identification of microbial mixtures.

Information about the possibility of identifying only major pathogens has been added to the manuscript.

10) L. 432 Chapter on ProteinChip Arrays.

Could you explain more in details this system?

RE: Thank You a lot for your remarks. More details about this system has been added to manuscript.

The ProteinChip technique is a de novo approach to protein discovery where prior knowledge of specific proteins is not required. The essential elements of the described technology are ProteinChip arrays, ProteinChip reader and dedicated software. Biological samples such as cell lysates, extracts or body fluids are applied to the ProteinChip Array, which allows proteins to bind to the surface based on chromatographic properties or specially designed biological affinity. Unbound molecules are flushed out, and proteins retained on the surface of the template are analyzed and detected using SELDI-TOF MS and the ProteinChip Reader. The obtained MS spectra are compared using differential protein mapping techniques, where the relative expression levels of specific molecular weights are compared using statistical and bioinformatics methods.

11) L. 533 with the sensitivity and specificity of 96.7% (Mycobacterium tuberculosis) and 91.7% of (non-tuberculous mycobacterial species).

Are both sensitivity and specify the same in both cases?

RE:There has been a misunderstanding of the source text by us. The information in this sentence has been corrected. Sensitivity and specificity were different and amounted to 96.7% and 91.7%, respectively.

12) In the Chapter 2 MALDI-TOF MS principle is described in detail. I miss the same description for other methods or MALDI-TOF MS modifications as e.g. MALDI-FT-ICR, MALDI-FTMS, MALDI-TOF Microflex.

RE:This chapter describes the general and basic principles of MALDI-TOF MS, the matrices used and the modes of operation.Modifications of this technique, such as MALDI-FT-ICR, MALDI-FTMS, are described in later chapters and subsections. However, as suggested, the existence of such techniques has been added to this paragraph.

13) It is under question whether Table 1 (l. 272) is really necessary in this article and what is this information value of it. The given information could be mentioned within the appropriate paragraph. Instead of this table if appropriate the table summarizing and comparing different approaches to use MALDI-TOF MS to identify ATB resistance might be formed.

RE:As suggested, Table 1 has been removed from the manuscript.

14) L. 884 4.4 Genomic chapter. I miss the information of using MALDI-TOF MS for genotyping (Vogel et al., 2009) or detection of SNP responsible for the antibiotic resistance (Griffin and Smith, 2000;  Sauer, 2006; Ikryannikovaet al., 2008). Please involve also this information in your review.

Vogel N., Schiebel K., Humeny A. (2009): Technologies in the Whole-Genome Age: MALDI-TOF-Based Genotyping. Transfusion Medicine, 36, 253-262.

Griffin T.J., Smith L.M. (2000): Single-nucleotide polymorphism analysis by MALDI-TOF mass spectrometry. Trends in Biotechnology, 18, 77-84.

Sauer S. (2006): Typing of single nucleotide polymorphisms by MALDI mass spectrometry: Principles and diagnostic applications. ClinicaChimicaActa, 363, 95-105.

Ikryannikova L.N., Shitikov E.A., Zhivankova D.G., Ilina E.N., Edelstein M.V., Govorun V.M. (2008): A MALDI TOF MS-based minisequencing method for rapid detection of TEM-type extended-spectrum beta-lactamases in clinical strains of Enterobacteriaceae. Journal of Microbiological Methods, 75, 385-391.

RE:Thank You for your remarks. The chapter has been supplemented with the proposed references.

The usage of English requires a minor revision in some stylistic parts. There are used some incorrect structures (allow for, allows the testing – l. 105, using “and” as “by how” – l. 87, he used instead of they used– l. 237) or some sentences are difficult to understand (l. 14-15, 275).

The formal side needs also to be improved in some cases as there are  some typing error (l. 440, 478, 510, 592, 87-89: a repeated sentence), missing gaps after final dots (e.g. l. 57, 108, 229, 645), redundant gaps within sentences (e.g. l. 135, 218,  concatenated words (e.g. l. 149, 228, 256, 385, 407, 602, 604) or missing letters (l. 41, 315, 472, 478) or missing italics (l. 34, 240, 426).

Thank You very much for your critical review. It was very useful in the correction of our manuscript. Identification of weak points throughout the text has helped us to increase the value of our paper. All comments and changes suggested by Reviewers have been incorporated into the manuscript. Once again, Thank You very much for your help.

Reviewer 3 Report

Janiszewska et al. have written an extensive review on the use of MALDI-TOF-related approaches to identify bacteria and test their antimicrobial susceptibility. The topic is interesting and in constant evolution and therefore deserves a good review. The review is overall well structured, and well written, except for specific parts (see below). Main strength are the parts dealing with technical and technological subjects. The English language needs extra attention, as too many strange sentences, grammatical errors, spelling mistakes etc are present. My major comments are:

1. I would like to see more information on advantages and disadvantages of the discribed techniques, especially in relation to the real life application. Sometimes techniques are described that are not practical and are also not used in diagnostics for that reason, but for which this is not clear in the text.

2. As the importance and possibilities of artificial intelligence and machine learning in the analysis and interpretation of MALDI-TOF related data has been described already for both bacterial diagnostics (including strain typing) and antimicrobial susceptibility testing, it is somewhat disappointing that this is not discussed at all. The importance of machine learning will probably only increase in the future, so the review will benefit of having this topic on board.

3. The authors should take care not to spent too much time on introductory paragraphs on topics that are not directly related to the MALDI-TOF techniques, for example on the issue of antimicrobial resistance. The reader needs to understand that this topic is important and that MALDI-TOF can play a role in this topic, but extended (yet still not complete and not entirely correct) discussion of the topic is not desirable. It is also quite clear that this is not the main topic of interest of the authors, as many mistakes are present.

4. Please check (very) recent literature. I have the impression literature of the last few years is somewhat underrepresented.

Specific remarks (some examples, non-limitative list):

Line 41: I would use "Salmonella", not "Salmonella enterica"

Line 50: not clear what last part of sentence means

Line 216: 10(5)-10(6) might be closer to the real-time limit of detection? 

Entire manuscript: please make sure to always explain abbreviations at first use (for example, but not limited to PMF, GPR, FTIR, Mtb, NTM, MLST...) and to use the correct explanation (for example, but not limited to MSP: Main Spectrum Profile; MRSA: methicillin (not metacycline) resistant Staphylococcus aureus; DOT-MGA: direct on target microdroplet growth assay...)

Line 263: "called"?

Line 268: "second system"; please be more specific which system

Line 315: This approachin?

Line 324: transport = transfer?

Line 440: "white range"? wide range?

Line 463-465: sentence reads odd

Line 466: not clear what "the units" are and relation with previous sentence not clear

Line 477: wasdemonstrated

Line 496-497: "is used" order of words in sentence is not logic

Line 507-508: sentence reads odd

Line 567-569: commercial extraction kit (Bruker) is available now and use of lipids is integrated in new software (lipidart) of Bruker. I do not know whether lipid analysis is also possible in platforms of other companies. Best to check latest information.

Line 604: detectcondensates

Line 631: nonocultures?

Line 748-783, including table 1: This part is not up to standards (see also major remarks) and should be shortened considerably. It is advised to get help from a bacteriologist with experience in this area to help with this part. Please omit the table, because it has no added value in this paper. Some examples: MRSA: methicillin (not metacycline) resistant Staphylococcus aureus. Disk diffusion is often used, but is not the gold standard technique, often this is broth microdilution; E-test is a commecrial name, best to use "gradient strip"; antimicrobial susceptibility testing based on genome analysis is promising, but is still dificult to interpret according to the clinical criterion. It is easier to interprete using the genetic or microbiological (epidemiological) criterion, but clinical relevance is not always clear.

Line 793-799: use of PSM in identifying MRSA is not very reliable (see for example Paskova et al., 2020 and Schuster et al., 2018): it has high specificity, but low sensitivity and is therefore also no longer used in commercial software. Best to add this information. In addition, there are other examples of the subtype module, like for example Bacteroides fragilis cfia detection and BlaKPC detection in E. coli and K. pneumoniae.

Line 800-814: this technique is not very practical, best to discuss. This also resulted in the MBT-ASTRA technique

Line 821: "Ceyssens misspelled

Line 845 and following: see also new software (Bruker) that is allowing to detect colistin resistance based on lipid analysis in negative ion mode

Line 862: colistin

Line 875-879: repetition from lines 815-824

Line 880-883: see more recent studies + also available for cephalosporins and carbapenems

Line 885-889: please discuss how practical this technique is and whether you expect this to be used in diagnostics soon or on longer term or not at all + why

Line 909-919: Not clear how relevant this paragraph is, also because English language is not up to standards

Line Line 923: "and identification"?

Line 958-960: is this truly the case? Please discuss practical issues

Author Response

Janiszewska et al. have written an extensive review on the use of MALDI-TOF-related approaches to identify bacteria and test their antimicrobial susceptibility. The topic is interesting and in constant evolution and therefore deserves a good review. The review is overall well structured, and well written, except for specific parts (see below). Main strength are the parts dealing with technical and technological subjects. The English language needs extra attention, as too many strange sentences, grammatical errors, spelling mistakes etc are present. My major comments are:

  1. I would like to see more information on advantages and disadvantages of the discribed techniques, especially in relation to the real life application. Sometimes techniques are described that are not practical and are also not used in diagnostics for that reason, but for which this is not clear in the text.

RE:Thank You a for your remarks. A description of the advantages and disadvantages of the MALDI-TOF MS technique has been added to the manuscript.

  1. As the importance and possibilities of artificial intelligence and machine learning in the analysis and interpretation of MALDI-TOF related data has been described already for both bacterial diagnostics (including strain typing) and antimicrobial susceptibility testing, it is somewhat disappointing that this is not discussed at all. The importance of machine learning will probably only increase in the future, so the review will benefit of having this topic on board.

RE:I agree that machine learners is a very important point.Therefore, information on the possibility of using machine learning to identify microorganisms using the MALDI-TOF MS technique was added to the manuscript.

  1. The authors should take care not to spent too much time on introductory paragraphs on topics that are not directly related to the MALDI-TOF techniques, for example on the issue of antimicrobial resistance. The reader needs to understand that this topic is important and that MALDI-TOF can play a role in this topic, but extended (yet still not complete and not entirely correct) discussion of the topic is not desirable. It is also quite clear that this is not the main topic of interest of the authors, as many mistakes are present.
  2. Please check (very) recent literature. I have the impression literature of the last few years is somewhat underrepresented.

Specific remarks (some examples, non-limitative list):

Line 41: I would use "Salmonella", not "Salmonella enterica"

RE: It was changes

Line 50: not clear what last part of sentence means

Line 216: 10(5)-10(6) might be closer to the real-time limit of detection? 

RE:It is 10(5) -10(6) bacterial cells/spot

Entire manuscript: please make sure to always explain abbreviations at first use (for example, but not limited to PMF, GPR, FTIR, Mtb, NTM, MLST...) and to use the correct explanation (for example, but not limited to MSP: Main Spectrum Profile; MRSA: methicillin (not metacycline) resistant Staphylococcus aureus; DOT-MGA: direct on target microdroplet growth assay...)

RE:Thank You a lot for your remarks. All suggested changes have been incorporated into the manuscript.

Line 263: "called"?

RE: The statement“…. calls Main Spectra Library (MSP).”has been changed to. “…referred to as the Main Spectra Library (MSP).”

Line 268: "second system"; please be more specific which system

RE: “second system” has been changed to “the VITEK®MS system”

Line 315: This approachin?

RE:It's an unfortunate combination of words.The words "approachin" have been changed to "approach in"

Line 324: transport = transfer?

RE: It should be “transfer”. It has been changed in text.

Line 440: "white range"? wide range?

RE: It has been changes to “wide range”

Line 463-465: sentence reads odd

RE:The sentence has been redrafted.

Line 466: not clear what "the units" are and relation with previous sentence not clear

RE: This sentence was removed because, upon careful reflection, it was inconsistent with the entire paragraph.

Line 477: wasdemonstrated

RE: It has been changed to “was demonstrated”

Line 496-497: "is used" order of words in sentence is not logic

Re: The sentence has been rewritten.

Line 507-508: sentence reads odd

RE:The sentence has been redrafted.

Line 567-569: commercial extraction kit (Bruker) is available now and use of lipids is integrated in new software (lipidart) of Bruker. I do not know whether lipid analysis is also possible in platforms of other companies. Best to check latest information.

RE:Information about lipid analysis software has been added to the paragraph

Line 604: detectcondensates

RE: It has been changed to “detected contensates”

Line 631: nonocultures?

RE: It has been changed to “monocultures”.

Line 748-783, including table 1: This part is not up to standards (see also major remarks) and should be shortened considerably. It is advised to get help from a bacteriologist with experience in this area to help with this part. Please omit the table, because it has no added value in this paper. Some examples: MRSA: methicillin (not metacycline) resistant Staphylococcus aureus. Disk diffusion is often used, but is not the gold standard technique, often this is broth microdilution; E-test is a commecrial name, best to use "gradient strip"; antimicrobial susceptibility testing based on genome analysis is promising, but is still dificult to interpret according to the clinical criterion. It is easier to interprete using the genetic or microbiological (epidemiological) criterion, but clinical relevance is not always clear.

RE:As suggested, Table 1 has been removed from the manuscript.

Line 793-799: use of PSM in identifying MRSA is not very reliable (see for example Paskova et al., 2020 and Schuster et al., 2018): it has high specificity, but low sensitivity and is therefore also no longer used in commercial software. Best to add this information. In addition, there are other examples of the subtype module, like for example Bacteroidesfragiliscfia detection and BlaKPC detection in E. coli and K. pneumoniae.

RE:Thank You a lot for your remarks. Recommended information has been added to manuscript.

Line 800-814: this technique is not very practical, best to discuss. This also resulted in the MBT-ASTRA technique

RE: Thank You for remarks. More information has been added to this paragraph.

Line 821: "Ceyssens misspelled

RE: It has been corrected

Line 845 and following: see also new software (Bruker) that is allowing to detect colistin resistance based on lipid analysis in negative ion mode

RE: More information about software has been added to manuscript.

Line 862: colistin

RE: It has been corrected

Line 875-879: repetition from lines 815-824

RE: The paragraph 4.3 has been redrafted.

Line 880-883: see more recent studies + also available for cephalosporins and carbapenems

RE: More information has been added to manuscript

Line 885-889: please discuss how practical this technique is and whether you expect this to be used in diagnostics soon or on longer term or not at all + why

RE: More information has been added to manuscript

Line 909-919: Not clear how relevant this paragraph is, also because English language is not up to standards

RE:Thank You a lot for your remarks. The English language in this paragraph and throughout the text has been improved. The purpose of this paragraph was to briefly describe where the increased resistance to antibiotics appears in the bacterial biofilm.

Line 923: "and identification"?

RE:This sentence has been transformed and clarified.

Line 958-960: is this truly the case? Please discuss practical issues

RE:This sentence has been reworded and is a summary to the entire described subsection.

Thank You very much for your critical review. It was very useful in the correction of our manuscript. Identification of weak points throughout the text has helped us to increase the value of our paper. All comments and changes suggested by Reviewers have been incorporated into the manuscript. Once again,Thank You very muchfor your help.